



# Evaluation of three new surface irrigation parameterizations in the WRF-ARW v3.8.1 model: the Po Valley (Italy) case study

Arianna Valmassoi[1,2], Jimy Dudhia[2], Silvana Di Sabatino[3], and Francesco Pilla[1]

[1]School of Architecture, Planning and Environmental Policy, University College Dublin, UCD Richview, Dublin, Ireland
[2]National Center for Atmospheric Research, Boulder, Colorado, United States
[3]Department of Physics and Astronomy, University of Bologna, Bologna, Italy

**Correspondence:** Arianna Valmassoi (arianna.valmassoi@ucdconnect.ie)

**Abstract.** Irrigation is one of the land managements that can affect the local climate. Recent literature shows that it affects mostly the near-surface variables and it is associated with an irrigation cooling effect. However, there is no common parameterization that also accounts for a realistic water amount, and these factors could be ascribed as causes of different impacts found in previous studies. This work aims to develop three new surface irrigation parameterizations within the WRF-ARW model (V3.8.1) that consider different evaporative processes. The parameterizations are tested on one of the regions where global studies disagree on the signal of irrigation: the Mediterranean area, and in particular the Po Valley. Three sets of experiments are performed using the same irrigation water amount of 5.7 mm/d, derived from Eurostat data. Two complementary validations are performed for July 2015: monthly mean, minimum and maximum temperature with ground stations and potential evapotranspiration with the MODIS product. All tests show that both mean and maximum temperature, as well as potential evapotranspiration, simulated fields approximate better the measures when using the irrigation parameterizations. This study addresses the sensitivity of the results to the parameterizations' human-decision assumptions: start time, length and frequency. The main impact of irrigation on surface variables such as soil moisture is due to the parameterization choice itself, rather than the timing. Moreover, on average, the atmosphere and soil variables are not very sensitive to the parameterizations assumptions for realistic timing and length.

## 1 Introduction

Irrigation has a crucial role in increasing the food production: while less than 20% of cultivated land is irrigated, it accounts for 40% of the global agricultural output (Bin Abdullah, 2006; Siebert and Döll, 2010). Irrigation is also responsible of 70% of the global water withdrawal and 80-90% of the consumption (Jägermeyr et al., 2015). In the context of increase in population and reaching sustainable living, the food production must increase to both sustain the current levels and ensure a fair distribution (Bin Abdullah, 2006). However, only to expand the arable land is an unlikely solution as the loss rate to urbanization, salinization and desertification is already faster than the addition one (Nair et al., 2013). Moreover, in the contest of the rapidly changing climate a shift in productions and cutivars has been already observed throughout the globe (IPCC, 2014; Wada et al., 2013; Lobell et al., 2008b; Zampieri et al., 2019, e.g). Anthropogenic influence on local climate is not only related to greenhouses emissions or changing in land cover, but also due to the land management practices. The practice that has the largest





impact is irrigation (Kueppers et al., 2007; Sacks et al., 2009; Wei et al., 2013). This is extensively used in semi-arid regions (Sridhar, 2013), such as the Mediterranean region (Giorgi and Lionello, 2008), and particularly during the summer growing period, whenever possible.

Recent literature shows that irrigation mostly affects near-surface atmospheric parameters, such as air temperature (Kueppers et al., 2007; Lobell et al., 2008a; Boucher et al., 2004; Sacks et al., 2009; Aegerter et al., 2017; Ozdogan et al., 2009; Sorooshian

et al., 2014, e.g.). The majority of the studies found that irrigation has a local cooling effect which does not clearly impact the global annual scale (Sacks et al., 2009; Boucher et al., 2004). Kueppers et al. (2007) found that the irrigation signal has a strong seasonal variability, with a maximum impact during the dry seasons of dry regions. In a global study, Lobell et al. (2006) found that the irrigation-induced cooling has a different magnitude depending on the analyzed region, varying from 8 K cooling to almost no effect, with a global cooling effect of 1.3 K. Another study by Boucher et al. (2004), with a global circulation model

with a simple bucket land surface model, found a global cooling of 0.05 K and a regional effect up to 0.8 K. Sacks et al. (2009) obtained similar results to Boucher et al. (2004), with also a regional cooling up to 0.8 K. However, in some specific regions, such as Southern Europe and India, the response to irrigation is less clear: Boucher et al. (2004) obtain an induced warming and Sacks et al. (2009) a cooling.This different cooling effect, in Sacks et al. (2009) is caused by the fact that surface temperature is more highly correlated with changes in downwelling radiation (linear regression: $r^2 = 0.49$), rather than changes in latent heat

($r^2 = 0.40$). However, it should be mentioned that a study with a later version of the CESM model by Thiery et al. (2017) found that the cooling is predominantly caused by an increase in evaporative fraction, with only a minor influence of reduced net radiation to the surface. This discrepancy in the causes was ascribed to the fact that, in the previous version of the atmospheric model (CAM3), convection was very sensitive to the surface latent heat changes (Thiery et al., 2017). Some of the regional studies did not find a significant change in the cloud cover (Kueppers et al., 2007, 2008; Sorooshian et al., 2011). In fact, most

of the variation is caused by the different surface energy balance partition between sensible and latent heat (Seneviratne and Stöckli, 2008) by increasing the supply of soil moisture available (Cook et al., 2010). Kueppers et al. (2007) found an inland irrigation-induced circulation pattern due to the contrast between the relatively cool, moist irrigated areas and adjacent warm, dry natural vegetation. Qian et al. (2013) found an impact of irrigation on the thermodynamic air masses properties, which might increase the probability of shallow cloud formation.

As mentioned, both global and regional modeling studies disagree on the magnitude and spatial pattern of these effects (Harding et al., 2015; Kueppers et al., 2007; Sacks et al., 2009; Lobell et al., 2006). Several studies ascribe the different impacts modeled to both the irrigation modeling (Leng et al., 2017) and the amount of water used Sorooshian et al. (2011); Wei et al. (2013); Sacks et al. (2009); Lobell et al. (2009). The methods vary depending on the study goal and model land surface process representation: for example, in Kueppers et al. (2007) the soil is maintained at the saturation point during the growing season

(also in Qian et al. (2013)); in Kioutsioukis et al. (2016) the irrigation is the amount of water requested by the difference between evapotranspiration and precipitation; Lobell et al. (2008a) keep the soil moisture at field capacity for the whole irrigated simulation period; in Sacks et al. (2009), the amount of water used for irrigation is applied to the surface directly, so it is given as input to the land surface model which partitions the additional water between evapotranspiration and runoff. While all these different ways to parameterize irrigation might be representative under the proper assumptions, the more realistic ones





(Sorooshian et al., 2011) are not yet implemented in the more widely used regional models, such as WRF. Moreover, most importantly the scheme proposed do not account explicitly for irrigation water amount as an input. These two, as Leng et al. (2017) point out, are crucial to assess, understand and quantify the irrigation signal at the regional scale, which is crucial to capture its local feature. It is to mention that CESM (in the CLM component), allows to calibrate the F-parameter to matching empirically the annual irrigation amount to the observed gross irrigation water usage for a specific period (Oleson et al., 2013;

Thiery et al., 2017; Leng et al., 2017). However, this irrigation implementation accounts only for evaporation from the soil, as it is applied by increasing the soil moisture.

    This study aims to provide a parameterization methodology for irrigation within a limited area model, which consider different evaporation processes. The mentioned parameterization will leave a choice for tuning parameters to account different

regions' irrigation management. In particular, we focus on one of the aforementioned regions where global circulation models have an uncertain irrigation impact: Southern Europe, the Mediterranean area. Irrigation methods and water used in the Mediterranean region depend on several factors such as cultivar type, climatic conditions and also water availability (Daccache et al., 2014). Due to the different conditions, only one sub region of the area is chosen: northern Italy and in particular the Po Valley (shown later in Fig.3). In this area, the majority of the water used to irrigate comes from surface water, the percentage

varies depending on the source from 71% (Fader et al. (2016)) to 95% (Ministero delle Politiche agricole alimentari e forestali (2009)). The remaining water is extracted from groundwater sources. Different methods are employed to irrigate the cultivars and for historical reasons the most common used is the "channel method": 52% of overall methods (Ministero delle Politiche agricole alimentari e forestali, 2009) or 61% (Fader et al., 2016). This method is common in this area due to its double function of irrigation and reclaiming. In fact, water is distributed by gravity-fed open channels and flows directly to the soil via siphons

or gated valves into furrows, basins or border strips (Van Alfen, 2014). The same channels are used to drain excess of water when necessary. The second most common method is irrigation through "sprinklers", both pivot and rain-like (Ministero delle Politiche agricole alimentari e forestali, 2009), for which the percentage varies from 24 to 25% depending on the source. Fader et al. (2016) includes also the "drip method", with a usage of 14% of the total methods, which is not included in the report from the Italian Ministry of Agriculture and Forest. Most of the water extraction for irrigation does not happen directly from the Po

River, but from the secondary rivers within the same basin (Ministero delle Politiche agricole alimentari e forestali, 2009). This study commences from previous studies' considerations about the impact of irrigation during dry growing seasons and the concern of common irrigation parameterization methods, which have tuning parameters. Firstly, irrigation parameterizations are developed for the widely-used Weather Research and Forecasting (WRF) model. The parameterizations are then tuned with the aforementioned irrigation methods currently deployed in the region. Consequently, the impact of irrigation on atmospheric

and soil components is discussed for the chosen area and simulation period.





## 2 Irrigation Parameterization Development

Irrigation processes are complex since they involve both human decision component and physical forcing. The work here aims to develop and implement an approach that allows the model to account for both the human management dimension and the physical response to the forcing. As irrigation methods' definition differs when different geographical area are considered

(Leng et al., 2017), the study here is going to characterize the different parameterization with the efficiency. Efficiency usually relates to unwanted water losses which can occur both in the system transportation and in the application. The first part can be related to numerical weather prediction models only when the transportation is performed through open channels, which leads to water loss due to evaporation. To account for such component, the model must have the capability of representing river processes, which WRF has not. Therefore, the second component of the efficiency, the water loss in the application, is consid-

ered for these parameterizations (similar to Leng et al. (2017)). As previous studies pointed out, depending on the irrigation techniques different physical processes has to be accounted for (Bavi et al., 2009; Uddin et al., 2010; Brouwer et al., 1990). For example, the sprinkler system looses water due to droplets evaporation and drift, as well as vegetation interception (Uddin et al., 2010; Brouwer et al., 1990). However, depending on the geographical area the techniques themselves varies (Leng et al., 2017). In fact, for some regions sprinkler are associated with systems that apply the water right above the canopy, so the water

loss due the droplet atmospheric processes is minimal. Other regions' most used sprinkler system are the centre-pivot, which might need to consider droplet processes if the irrigated field radius is big enough. Therefore, to account for different regional interpretations, the parameterizations are defined based on the processes considered. Nonetheless, specific names are used for simplicity to differentiate the schemes within the model itself and for testing. However, they are not necessarily intended as resembling techniques used in real cases.

The methods presented in the next paragraphs consider an increasing amount of evaporation processes, after the water leaves the irrigation system. In particular, the main process considered are represented in the scheme in Fig. 1. In this framework, it is to notice that the water is introduced from a source that is considered not connected to the current system. While the source withdrawal component was found to have a key role from the theoretical perspective (Leng et al., 2017), WRF has not the capability of reproducing the surface water dynamic.

The implementation of the schematics within the WRF model are described in detail in the following part. The naming convention resemble the actual techniques for Opt.1 and 3, respectively with CHANNEL and SPRINKLER. To avoid misrepresentation with Opt.2, the naming is chosen to recall the specifics as **DR**ip on **l**eaves as **P**recipitation or DRlP.

### 2.1 Option 1: CHANNEL

This method accounts only for evaporation from the soil and water at the surface, and the equations describing it are defined

by the land surface model chosen. The irrigation water is added in the rain variable, and it is given to the land model as an input parameter. Therefore, such modification does not affect the atmospheric parameterizations and equations directly, but only indirectly.

For simplicity, the water used for the irrigation is defined, from an input, as an average daily amount expressed in millimeters





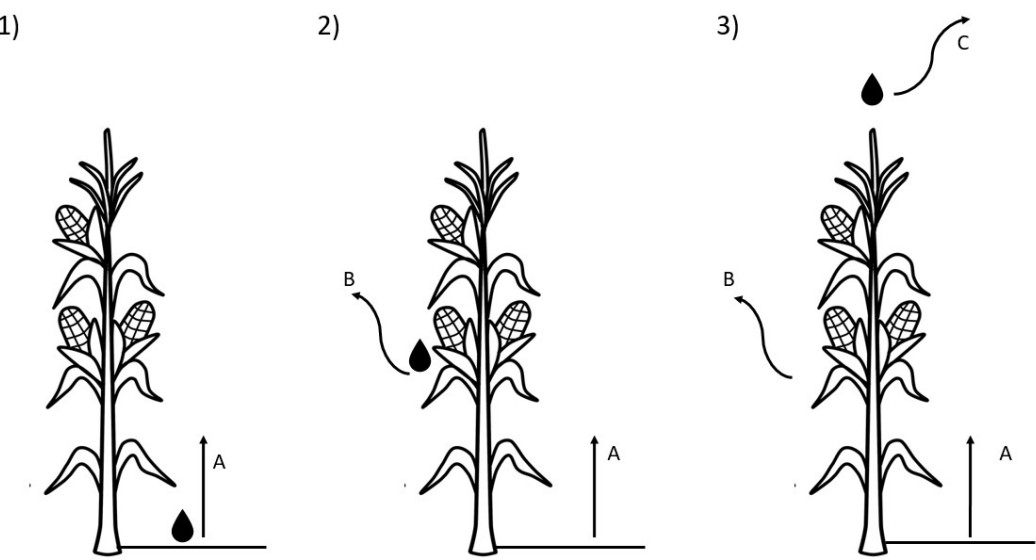

**Figure 1.** Irrigation schemes (1-3) with increasing evaporative processes considered (A-C): A is the evaporation from the water at the soil level, B is the canopy interception and C is the drop evaporation and drift. This framework accounts only for surface water application, and not sub-surface.

(*irr_daily_amount*: $V_I$ in mm/d, which is then converted in mm/s). Moreover, irrigation is set to start at the UTC-time defined from *irr_start_hour* and it is partitioned equally during the consecutive *irr_num_hours*-hours ($h_I$, converted in seconds). To conform it with precipitation, $W_I$ [mm/s] is expressed as:

$$W_I = \frac{V_I}{h_I}\Delta T_I \tag{1}$$

The obtained amount of water $W_I$ is then integrated in the model timestep. $\Delta T_I$ is the irrigation frequency, expressed as absolute number of days, which accounts for not-daily cases. This variable is used to compensate the water quantity during the period when the irrigation method is not applied. This is the easiest way to have a fixed total amount of water for a simulation, which considers different irrigation frequencies. When the model is in the irrigation interval defined by the start hour, $h_I$ and $\Delta T_I$, $W_I$ is constant and defined as Eq. 1. Outside this interval, $W_I$ is set to zero. A start and end day for irrigation can be defined using the Julian day calendar representation.

The evaporation processes that irrigation water undergoes are defined only by the land surface scheme chosen. However, in this method, the processes determined by the canopy interception are not considered. Therefore, the water accumulated on the canopy is imposed to zero when irrigation is activated.





## 2.2 Option 2 : DRIP

This representation allows considering the water interception from the canopy and the leaves. In particular, it considers the water as applied right above the canopy. Once on the canopy, the water can undergo evaporation from the leaves and/or drip to the ground. The specific processes included in the representation of water intercepted by the canopy depends on the land

surface scheme itself.

This scheme uses the same approach to include irrigation, i.e. via the surface precipitation, as the previous option. However, differently from the previous one, the water undergoes all rain processes related to canopy water balances. For example, if the land surface scheme allows a partition of the rain between interception and dripping, Option 2 will include the interception, but Option 1 will not.

## 2.3 Option 3: SPRINKLER

This option includes the droplet processes, which in WRF are described in the micro-physics schemes. Here, the irrigation is considered as water sprayed into the lowest part of the atmosphere, namely the first full model level above the ground. The specific processes that the irrigated water undergo in the micro-physics depend on the choice of the scheme itself. However, all includes the evaporation of the rain droplets, as well as advection.

This method assumes a static input of irrigated water directly into the rain water mixing ratio (a field that is input in all schemes) as mass within the volume-grid point. This avoids any assumptions such as the falling speed or droplet size distribution. Therefore, the new rain water mixing ratio ($Q_r$ [kg/kg]) includes the irrigation in the lowest model layer as:

$$Q_r = Q_r + Q_I \tag{2}$$

The total grid point mass rate of water ($m_I$ [kg/m3.s]) added to the lowest mass level ($\Delta z_{ks}$ [m]), per cubic meter, is:

$$m_I = \frac{W_I}{\Delta z_{ks}} \tag{3}$$

where, $W_I$ has already been defined in Eq. 1. If Eq.3 is divided by the lowest mass level air density ($\rho(t)_{i,j,ks}$ mass per cubic

meter), it leads to the irrigation mixing ratio ($Q_I$, in [$\frac{kg}{kg \cdot s}$] ):

$$Q_I(t)_{i,j,ks} = \frac{W_I}{\Delta z_{ks} \, \rho(t)_{i,j,ks}} \tag{4}$$

This value obtained is integrated on the microphysics timestep, so it becomes kg/kg, before adding it to the rain mixing ratio. With this option, the microphysic scheme describes the evaporation processes that irrigation water drops undergoes exactly as the rain droplets. After this, the irrigation water enters the model workflow as part of the microphysic precipitation field. Therefore, it is subject to the evaporation processes from canopy interception and the soil as they are described in the chosen

schemes.





## 2.4 Irrigation Mask Field

To increase the precision of where the irrigation takes place, the FAO's AQUASTAT database (Siebert et al., 2013) is used. This global gridded dataset combines national level census data of agricultural water usage for areas equipped for irrigation, with a resolution of $0.0833°$ (around 9.24 km at mid latitudes). The dataset is included in the "geogrid" WPS preprocessing feature

as an optional field, giving the percentage of irrigated land within the volume grid cell. This allows the field to be interpolated consistently with all the other geographical ones to the chosen grid resolution. The water applied for irrigation, as described in the previous methods, is weighted on the percentage of irrigated land within the grid point.

## 2.5 Irrigation Frequency Greater than Daily

As previously seen, the irrigation frequency can be different than daily. This choice leads to a different behavior than having a

sub-grid variability of irrigation, which would result in a lesser water amount used per grid point. In fact, it allows investigation of the transition of the soil between intense irrigation states and days without any.

We can define two regimes accordingly to the frequency: synchronous or not synchronous. This allows to consider that on multi-daily frequency, the whole area might not be irrigated at the same time, but on different days within the period. The use of this option leads to the possibility of having different spatial patterns when the irrigation is not synchronous. In the case of

the synchronous irrigation, the chosen method is activated for the whole domain with the timing chosen by the combination of $\Delta T_I$, *irr_start_hour*, *irr_num_hours*, and *irr_start_julianday*. In fact, the active day has to be a multiple of $\Delta T_I$ counting from the irrigation starting day. In the case of non non-synchronous irrigation, where grid cells have the same frequency but different phases, the activation field is defined as a static random field. This uses the Fortran RANDOM_SEED function to create a repeatable random array that is given to calculate the activation field with the RANDOM_NUMBER Fortran function.

However, this option does not ensure a reproducibility of the random field across different compilers. To the current method, an additional one reproducible across different compilers is given as an option.

## 3 Methods

### 3.1 Model Settings

The numerical weather prediction model used is the non-hydrostatic Weather Research and Forecast (WRF) model V3.8.1

(Skamarock et al., 2008). In particular, the Advanced Research WRF (ARW, or WRF-ARW) dynamical solver it is used for this study(Skamarock et al., 2008), hereafter when referring to the model used it is implied that is WRF with the ARW solver. In the study, WRF is used to test the parameterizations first for a 16 days period, and then for a longer one. Therefore, it is important that the domain is correctly forced by the boundary conditions in order to have a long continuous run. The forcing by the boundaries, that the domain of interest is sufficiently close to the outer domain boundary, is used to keep the model on

the right path, so the non-linearities intrinsic in the fluid dynamics and physics are constrained and the model does not diverge much from analyses.



The initial and boundary conditions for atmosphere and soil are chosen from different model products. For the atmosphere, ERA-Interim is used because it is a state of the art of atmospheric reanalysis. In particular, note that ERA-interim is an ECMWF global atmospheric multi-decade reanalysis product which uses a 6 hours 4D-Var data assimilation system with both ground and upper atmosphere data sources (Dee et al., 2011). Differently from the previously used NCEP data, ERA-Interim has a spatial resolution of approximately 80 km (around 0.75 degrees; it is a T255 spectral grid) on 60 vertical levels from the surface up to 0.1 hPa (Dee et al., 2011). This allows nesting directly from the boundary conditions to a 15 km resolution domain. As for the soil initial conditions, the GFS 0.25° product (cis, 2015) is used because of its similarity to WRF's Noah LSM in the parameterization and soil level discretization (Ek, 2003). This allows a more consistent initial condition for the soil layer temperatures and moisture.

Moreover, the MODIS 15 arc-second (around 450 meters resolution) dataset is used for the land-category definition in the studied area. This is the most accurate dataset available for this region. As introduced in previous studies, WRF has the capability to nest multiple domains in the same run, reducing the total computational time and improving local climate representation. The configuration chosen is centered on the Po Valley, and it is shown in left of Fig. 2. The outer-most domain has a 15 km

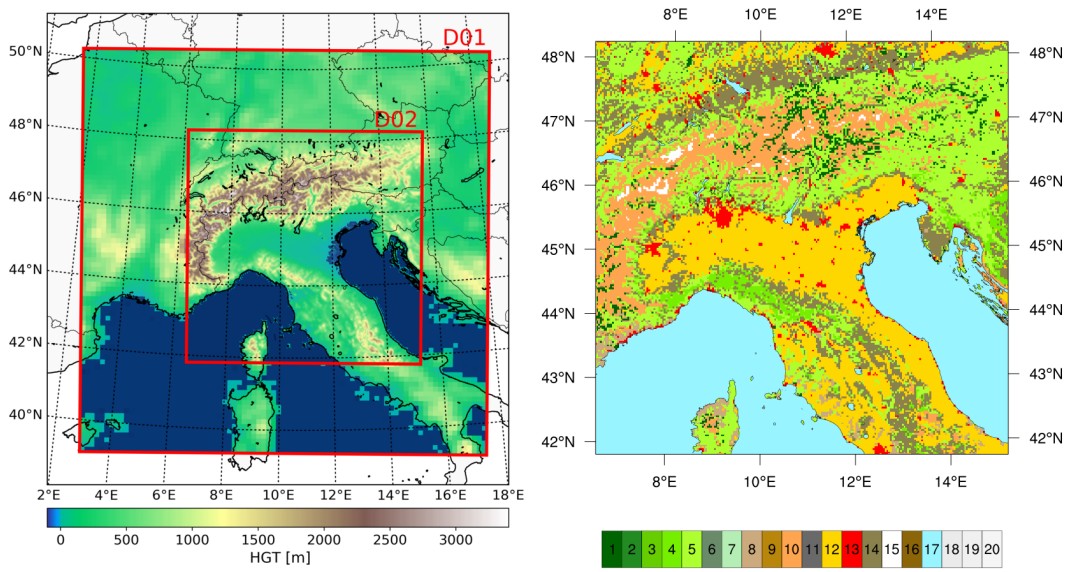

**Figure 2.** The left figure show both domains: the outer (D01) with 15 km grid resolution, and the inner (D02) with 3 km. The right figure shows the land use categories used: 1-5 represent different forest type, 12 croplands and 13 is built-up ( for more information about the specific categories refer to Skamarock et al. (2008)).

resolution and covers part of the northern Mediterranean area. The nested domain, called $D02$, has 3 km resolution and covers part of Italy and the Alpine region. The inclusion of the Alpine region is forced by the presence of such a complex terrain as the Alps, which can cause the model to misbehave if it intersects a domain boundary. Therefore, to better represent the terrain, and the atmospheric behavior, the Alpine region is included in the finer domain.



Given the pre-processing choices, the parameterizations used are presented. The RRTMG radiation scheme for both long wave

and short wave radiation is used since it is commonly used in this area of interest (Mooney et al., 2013; Stergiou et al., 2017). The Newer Tiedtke cumulus scheme is used for the outer domain that needs a convective parameterization; this parameterization is similar to the ECMWF cumulus scheme operationally used in the model (Zhang and Wang, 2017). This allows us to have a consistent cumulus parameterization with the boundary and initial conditions. The single-moment 6-classes (WSM6) microphysics scheme (Hong et al., 2005) is used here due to its lesser computational cost with respect to others that have the

same complexity. As in previous WRF studies, the YSU boundary layer parameterization is also used here (Hong et al., 2006). As mentioned before, the land surface model used for this study is Noah, which is the same model, but different version of the one used in GFS (Mitchell et al., 2005; Tewari et al., 2004). The timestep used for all scheme is the same as the model timestep, which for the outer domain is 60 seconds and follows the 1:5 ratio for the inner domain, so 12 seconds.

The irrigation mask derived from the FAO dataset, for Po Valley in the high resolution domain (D02) is shown in Fig. 3. As it

**Figure 3.** Percentage of irrigated area after regridding for the Po Valley. The red box highlights the averaging area used in this work.

is possible to see, most of the western part of the Po Valley has more than 60% of the land irrigated. The eastern side has lower irrigated percentages.

### 3.2 Test Case: Summer 2015, Po Valley

As mentioned before, the impact of irrigation is greater in drier and warmer seasons, so that the irrigation signal is not masked by precipitation or larger scales systems. Summer 2015 was a particularly dry and warm season, with potential soil moisture

deficit due to winter precipitation anomalies. While June 2015 was an average month with respect to the period 1981-2010, July was exceptional ARPAE (2015a, b)). In fact, for the eastern part of the Po Valley the temperature maxima were 1.8$^o$C above the one measured in July 2003 during the famous heat wave (for more about 2003 heat wave refer to Della-Marta et al.





(2007) or García-Herrera et al. (2010)). July 2015 registered negligible precipitation on the eastern side of the Po Valley, and the return period associated with the experienced soil moisture deficit is between 20 and 50 years (ARPAE, 2015b).

In the presented work the different methods previously described are tested for the chosen period. For the chosen area, the methods are going to be related to the techniques defined by Ministero delle Politiche agricole alimentari e forestali (2009). The channel method in the Po Valley release the water onto the surface of the field, without interception of the vegetative canopy, so it resemble the Option 1 (CHANNEL, hereafter). Different irrigation sprinkler systems are used in the region, as both less efficient sprinkler guns are widely deployed as well as the more efficient rain-like (Ministero delle Politiche agricole

alimentari e forestali, 2009). As a matter of naming definition, the option 3 that represents the least efficient option is called SPRINKLER (as originally defined in Leng et al. (2017)) later in the study. In the current model setting, the lower full model level is about 10 meters thick. Option 2 is defined as the irrigation system with the water dripping over the leaves, and for brevity it will be called DRIP[1] hereafter. The terms here defined are not to be intended as an universal technique definition, but as a naming convention within this case study.

The irrigation water amount is derived for the area of interest of Fig. 3. The total amount of water used is $8.209 \cdot 10^{12}$ liters (Eurostat, 2013), which is distributed on $1.5505 \cdot 10^{10} \, m^2$. The area considered already accounts for the percentage of irrigated land within the grid-point as defined by FAO. To have a uniform temporal behavior for irrigation, it is assumed that it is applied every day from 15 May to 15 August, for a total period of 92 days. Therefore, the total amount of irrigation used in the region is $5.7 \, mm \cdot day^{-1}$. The total water amount used for irrigation through the experiments will be the same, since the water amount

is normalized (Eq.1).

To address the effect of irrigation on the local climate, several experiments are performed for different spatial resolutions and temporal periods. Even though the periods might be different, they all include at least part of July 2015. For averaging purposes, only runs that have the complete average period are included in the processes. To summarize the different features, all experiments are summarized in the Table 1. Each experiment is then described in detail below.

**Table 1.** Table with the experiments used in this paper: the main features of them are summarized here. For further explanations refer to the main body text. "various[a]" refers to the various settings that for simplicity reasons are described in Table 2.

| Name | simulated period | Resolution | Spin-up | Irrigation settings | | |
|---|---|---|---|---|---|---|
| [Acronym] | | [km] | [days] | Start [UTC] | length [hours] | $V_I$ [mm/d] |
| TR1 | 1-17 July | 15 | - | 5 | 3 | 5.7 |
| TR2 | 1-17 July | 3 | - | 5 | 3 | 5.7 |
| SR0-8 | 1-31 July | 15 | - | various[a] | various[a] | 5.7 |
| LR1 | 1 May - 31 July | 15 | 15 | 5 | 3 | 5.7 |
| LR2 | 1 May - 31 July | 3 | 15 | 5 | 3 | 5.7 |

---

[1]It is to remember that it does not resemble the actual drip irrigation method (defined as in (FAO, 1988)) and anyways such technique is not deployed in this area (Ministero delle Politiche agricole alimentari e forestali, 2009).





### 3.2.1 Test Run [TR]

This part of the experiment uses a subset of summer 2015, due to the high anomalies registered in the region: from 1 to 17 of July at 00 UTC. The water amount is then distributed every day from 05 UTC (7 AM local time) for 3 hours, only in the inner domain. The irrigation is applied throughout the whole simulation period, without a spin-up time, therefore the start and end day are not relevant.

Such a short period of simulation is used to test the scale dependency of the results. In fact, these settings are applied once to the outer domain (TR1) and once only to the inner domain (TR2). The TR1 is used to test the schemes at the 15 km resolution only, while TR2 has the schemes only in convection-permitting D02 domain.

### 3.2.2 Sensitivity with Coarse Domain [SR]

This part of the experiment is used to test the dependency of the results to the starting time and irrigation length. Due to computational constraints, the sensitivity study is done with the 15 km domain and for the July 2015 month. This ensures a high number of sensitivity members for different options. Table 2 summarizes the design of 9 different settings that are applied to all three parameterizations. With the number zero is indicated the chosen reference irrigation scenario, called "standard run".

**Table 2.** Table of number indexes used, e.g. CHAN4 is channel method fourth combination of the table here.

| Combination number | starting time [UTC] | length time [hours] | frequency [days] | phase |
|---|---|---|---|---|
| 0 | 5 | 3 | 1 | 0 |
| 1 | 17 | 3 | 1 | 0 |
| 2 | 12 | 3 | 1 | 0 |
| 3 | 5 | 1 | 1 | 0 |
| 4 | 5 | 5 | 1 | 0 |
| 5 | 17 | 1 | 1 | 0 |
| 6 | 17 | 5 | 1 | 0 |
| 7 | 5 | 3 | 3 | 2 |
| 8 | 5 | 3 | 7 | 2 |

There is also a control run with no irrigation. Therefore, the sensitivity has a total of 27 tests plus a common control simulation (CTRL SR). The starting time values are divided between early morning and late afternoon; one test is performed also for the middle of the day. More intuitive are the choices to irrigate either earlier in the morning or later in the afternoon: the water loss by evaporation and evapotranspiration is minimized, therefore the plant uptake is maximized. Combination 2 (start at 12 UTC) ensures that the representation of one of the least favorable irrigation conditions is also captured. In fact, during noon time the high temperatures in both soil and atmosphere are favorable to water evaporation.

These sensitivity settings are also used to test the non uniform temporal feature of the parameterization implementation, which

are highlighted by the second part part of table. In the first of these tests, irrigation is activated every three days (combination
number 7 of Tab.2) and the second every seven days (combination number 8). Here, only the random static field approximation
is tested since the configuration does not differ much from the pseudo-random one. As previously described, the frequency in
days determines the activation field. For clarity both activation fields are shown in Fig. 4. The values represent the number of

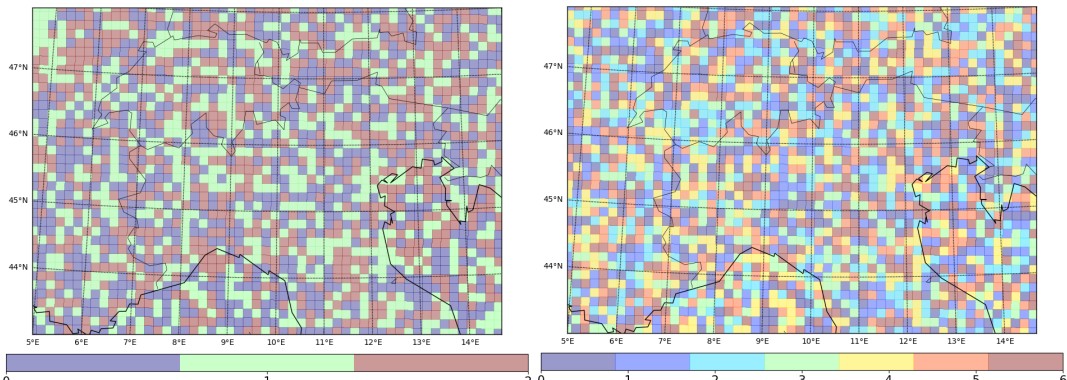

**Figure 4.** Activation field in days since the start of the sequence counting. Highlight of the Po Valley region.

the day within the sequence repetition in which the irrigation will be activated. For example, if a grid value is zero, it means

that irrigation will happen on the first day of the interval between irrigation times.

### 3.2.3  Long Run [LR]

This set of experiments is done to address the longer term influence of the developed irrigation parameters. The period simulated
started from 1 of May 2015 and end 31 of July 2015. This simulation set uses only the June and July months for the analysis,
as May is considered spin-up. This means that for the control, the soil moisture has a spin-up time of 31 days. However, in

the case of the irrigated runs, the first 15 days are without the schemes active, and then 16 days are for irrigation to reach
a new equilibrium. The water amount used is 5.7 mm/day, which is the same as all the other experiments. The long run
experiment has the so-called "standard configuration": every day from 05 UTC for 3 hours, which was also used for the Test
Run. The aforementioned settings are used for both a high resolution simulation (LR2) and a coarse one (LR1), for all three
parameterizations and a control run.

In this study, this experimental setting is used only to validate the parameterizations. More in depth analysis of the high
resolution results is out of the scope of this paper.

### 4  Validation

The validation of the proposed parameterizations inclusion within WRF consists of using stations' 2-meter temperature and
satellite potential evapotranspiration. The stations' data are from the regional weather services (ARPA, from the regions Emilia-





Romagna, Lombardia, Piemonte) and have an hourly frequency. From all the available stations, only the non-urban ones are used. The potential evapotranspiration is a product from the Moderate Resolution Imaging Spectroradiometer (MODIS) Terra dataset (Running et al., 2017).

### 4.1 Surface Network of Monitoring Weather Stations

Previous studies reported that irrigation affects the temperature and Kueppers et al. (2007) reported an impact on the maximum
diurnal temperature, but no clear effects on the minimum one. Therefore, this first part of the validation consists of comparing the model output of the high-resolution long run (LR2) to the stations' data. Of the three months in the LR2, only the last one is used for the validation: July 2015. As previous studies shows, dry months show the irrigation signal stronger (Kueppers et al., 2008; Leng et al., 2017). Therefore, the model performance is going to be affected more clearly by the parameterizations, so it helps to isolate the signal. As May is used as spin-up, and June is an average month, July is used. Differences are defined
as the model results minus the station data and called bias hereafter. A bi-linear interpolation is performed using the stations' coordinates to approximate the model gridded data to their locations. If in the interpolation the model land use category is not cropland, then the station is not used. This ensures that the model point results are not influenced by other land use physics, such as urban. Even though this should ensure that all stations and model points are actually in agricultural fields, the reality of the stations' location is different. This is especially true for the Arpa Lombardia stations, where not all are standard WMO
or representative of their surrounding environment (e.g. station 37, as later explained).

Moreover, to ensure that the stations have a sufficient number of data for the monthly average process, only stations with at least 80% of the hourly values are used. This process leaves with 44 stations out of the 62 downloaded originally. To understand the behavior of the biases, defined as the difference between the model run value for the location and the respective station, the mean, median and standard deviation are calculated. The results of this mean process are shown in Tab. 3. Two percentages are

**Table 3.** Indexes for the monthly values of the mean T2, the mean of the daily maximum and minimum temperature, for the valid stations: mean ($\overline{x}$), median ($\widetilde{x}$), standard deviation ($\sigma$), percentage of stations with positive bias ($\beta^+$) and percentage of stations ($\beta^*$) with a bias less than $|0.5|^oC$.

| | T2 | | | | | $T2_{max}$ | | | | | $T2_{min}$ | | | | |
| | $\overline{x}$ | $\widetilde{x}$ | $\sigma$ | $\beta^+$ | $\beta^*$ | $\overline{x}$ | $\widetilde{x}$ | $\sigma$ | $\beta^+$ | $\beta^*$ | $\overline{x}$ | $\widetilde{x}$ | $\sigma$ | $\beta^+$ | $\beta^*$ |
|---|---|---|---|---|---|---|---|---|---|---|---|---|---|---|---|
| CTRL | 0.75 | 0.50 | 0.88 | 82 % | 45 % | 1.46 | 1.21 | 0.94 | 95 % | 14 % | -0.59 | -1.11 | 1.63 | 27 % | 18 % |
| CHAN | -0.06 | -0.22 | 0.86 | 39 % | 41 % | -0.21 | -0.23 | 0.95 | 41 % | 41 % | -0.34 | -0.83 | 1.46 | 30 % | 18 % |
| SPRI | -0.20 | -0.40 | 0.87 | 30 % | 41 % | 0.11 | 0.15 | 0.99 | 59 % | 36 % | -0.52 | -1.10 | 1.56 | 27 % | 18 % |
| DRIP | -0.19 | -0.40 | 0.87 | 30 % | 43 % | -0.27 | -0.20 | 0.97 | 43 % | 43 % | -0.47 | -1.01 | 1.51 | 27 % | 18 % |

added to the aforementioned statistics indexes: the stations with positive bias and the ones with a bias less than $|0.5|^oC$.

In addition to Table 3, the biases are plotted with a shading of the IRRIGATION field of Fig.3 to visually understand the impact of the parameterizations implemented. This allows for visualization of the irrigation pattern and contextualizing the spatial influence of the changes in temperature caused by the parameterizations.

The first thing highlighted by Tab.3 is that irrigation affects the biases mostly as concerns the mean and maximum temperature,



but not the minimum temperature. This finding agrees with previous works' results, as Kueppers et al. (2007). From Tab.3, both the mean biases and the percentage of stations with positive bias are reduced significantly. On the other hand, the standard deviation of the biases is not strongly affected. All the methods lead to an over-decrease of the mean 2-meter height temperature, with more stations with a negative bias than a positive one ($\beta^+$), as Fig.5 left. Despite that, the number of stations with a

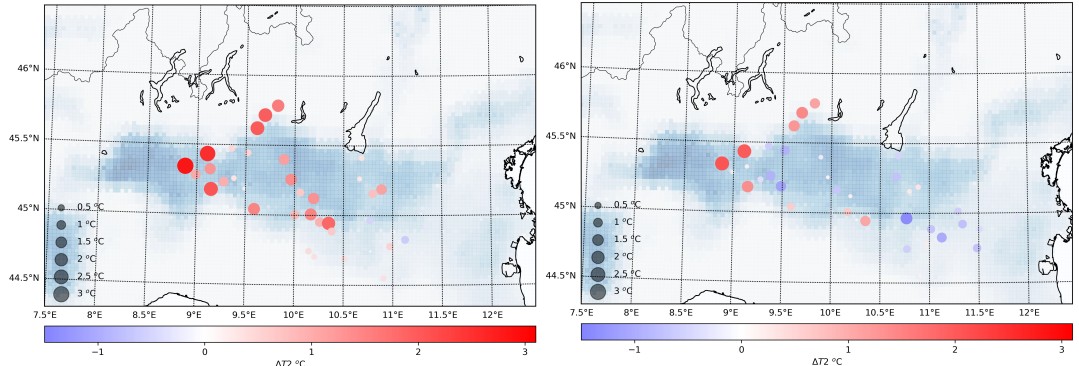

**Figure 5.** Monthly average for the 2-meter temperature differences between the control (left) or channel (Opt.1, right) run and the weather stations' location. This is for the July 2015 from the 3-month simulation (LR2). Both dot size and color represent the bias, and they are used combined to highlight high values.

bias between $\pm 0.5^oC$ ($\beta^*$) decrease only slightly compared to the control run, which agrees with Fig.5. Some stations show a
bias, up to $3^oC$, that is not strongly affected by the schemes, and they are the ARPA Lombardia stations previously mentioned. For example, the station 37 (Table A1) is located on the bridge above the Ticino river. Therefore, a strong bias is expected since the model does not have a water body in the area. The three stations (number 27,41 and 43 of Table A1) are in an Alpine valley, which can lead to a different set of model biases, such as the effect of steep terrain. Despite these external issues with the stations, the irrigation parameterization still improves the biases.

Maximum daily temperature is the quantity that shows the best performance improvement. In fact, all indexes report a significant improvement. The only exception is the standard deviation which is not affected by the use of the irrigation parameterization. In particular, the control run has 95% of the stations with a positive bias, and only 14% within $\pm 0.5^oC$ (Fig.6 top left). All the irrigation parameterizations $\beta^+$ and $\beta^*$ values are closer to more optimal values. Interestingly for the maximum daily temperature, with the irrigation parameterization the mean is similar to the medians, which was not the case for the
control run. It seems to improve the uniformity of the distribution of the biases, even though the irrigation field is not uniform. The CHANNEL and the DRIP parameterizations show similar spatial behaviors in the biases magnitude and distribution. The sprinkler scheme, instead, presents a bigger increase in the negative biases in highly irrigated areas, and a lesser decrease of the positive ones in area with low percentage of irrigated land. This behavior is expected due to the physical representation of the sprinkler, which directly affects the atmosphere. However, the increase of points with negative biases does not offset the
one with a positive ($\beta^+$).

The monthly minimum daily temperature is the quantity least affected by the irrigation scheme. In this case, the statistics in-



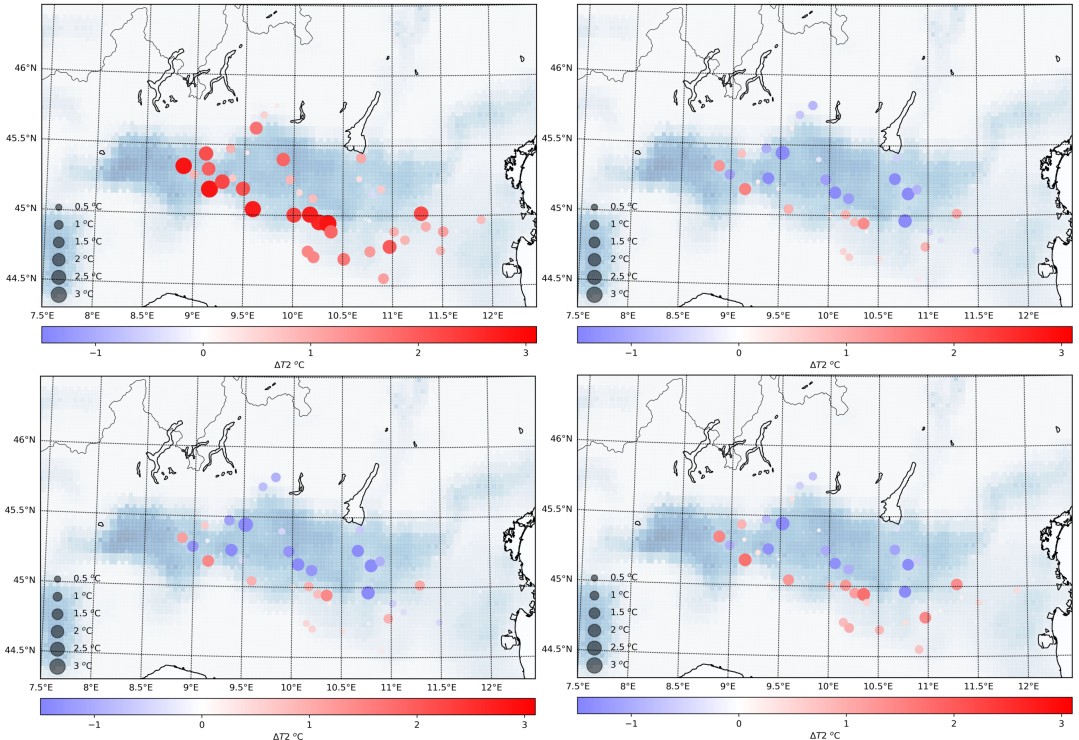

**Figure 6.** Monthly average for daily maximum 2-meter temperature difference between model run LR2 and the weather stations' location for the control run (top left), CHANNEL (Opt.1, top right), DRIP (Opt.2, bottom left) and SPRINKLER (Opt.3, bottom right)

dices show almost no variation for the bias distribution due to the irrigation parameterization. In fact, the underestimation ($\beta^+$) of the monthly minimum temperature does not change depending on the four runs. On the other hand, the mean and median values slightly improve with the irrigation schemes. The high standard deviation observed is caused by the ARPA Lombardia stations previously discussed, with a positive bias over $3^oC$ in Fig.7 (which are Station number 41,42 Table A1). In particular, the channel parameterization shows the bigger improvement of the negative biases in the southern part of the region observed in Fig. 7 (top left). All the schemes do not affect significantly the positive biases in the irrigated area. Moreover, the SPRINKLER and DRIP schemes have a similar impact on the biases (Fig. 7 bottom right and 7 bottom left).

### 4.2 Potential Evapotranspiration

The potential evapotranspiration can be considered as the evaporative demand from the atmosphere to the surface, as it is the maximum ability to evaporate under the assumption of a well-watered surface (Thornthwaite, 1948). As for satellite data, potential evapotranspiration is an indirect quantity, since it is derived from multiple measurements of satellite channels. Within the MODIS products there is also the evapotranspiration, which is the net effect between the evaporation demand and the availability. This could be a better quantity to estimate the effect of irrigation on the system. However, MODIS evapotranspiration



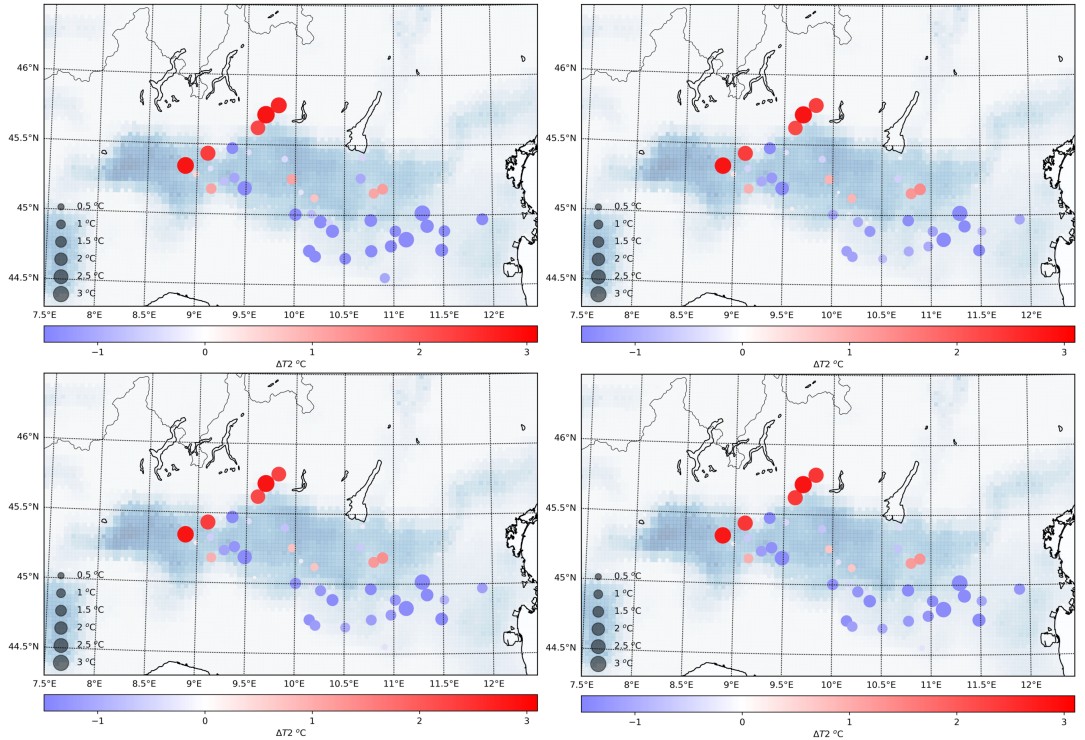

**Figure 7.** Monthly average for daily minimum 2-meter temperature difference between model run LR2 and the weather stations' location for
the control run (top left), CHANNEL (Opt.1, top right), DRIP (Opt.2, bottom left) and SPRINKLER (Opt.3, bottom right)

is the result of a daily algorithm that combines both satellite measures and atmospheric models, as well as surface parameter
assumptions (Running et al., 2017). Therefore, the assumptions of the evapotranspiration calculation makes the quantity not
ideal for validation purposes.

The potential evapotranspiration (PET) from MODIS is an 8-day accumulated product, with a 500 m resolution, which is finer
than the 15 km and 3 km resolution used for the models. Therefore, to compare such different scales, only the accumulated

values for the whole irrigated area of the Po Valley are considered. Due to the different temporal resolution between the data,
the potential evapotranspiration is summed for the whole July 2015 period. The process is applied to the sensitivity run (SR)
as well as the long run LR1 and LR2. The results of the process are shown in Fig. 8. The accumulated value obtained for the
MODIS data is aggregated with the control run of Fig.8 and it is about 243 mm. The measured value is from 33% to 17%
lower than PET from the control run. All irrigated runs show an improvement of the potential evapotranspiration, decreasing

the previous bias values to 23% and 12%. The highest improvement is observed in the 3-months simulations LR1 and LR2,
of which only the last month is used. The potential evapotranspiration in the long control run (CTRL LR1) is higher than the
one in the sensitivity control run (CTRL SR). In the case of the control run, SR and LR differ only for the spin-up time, as
SR starts on the $1^{st}$ of July and LR is the three months simulation. Therefore the evolution to the equilibrium of variables





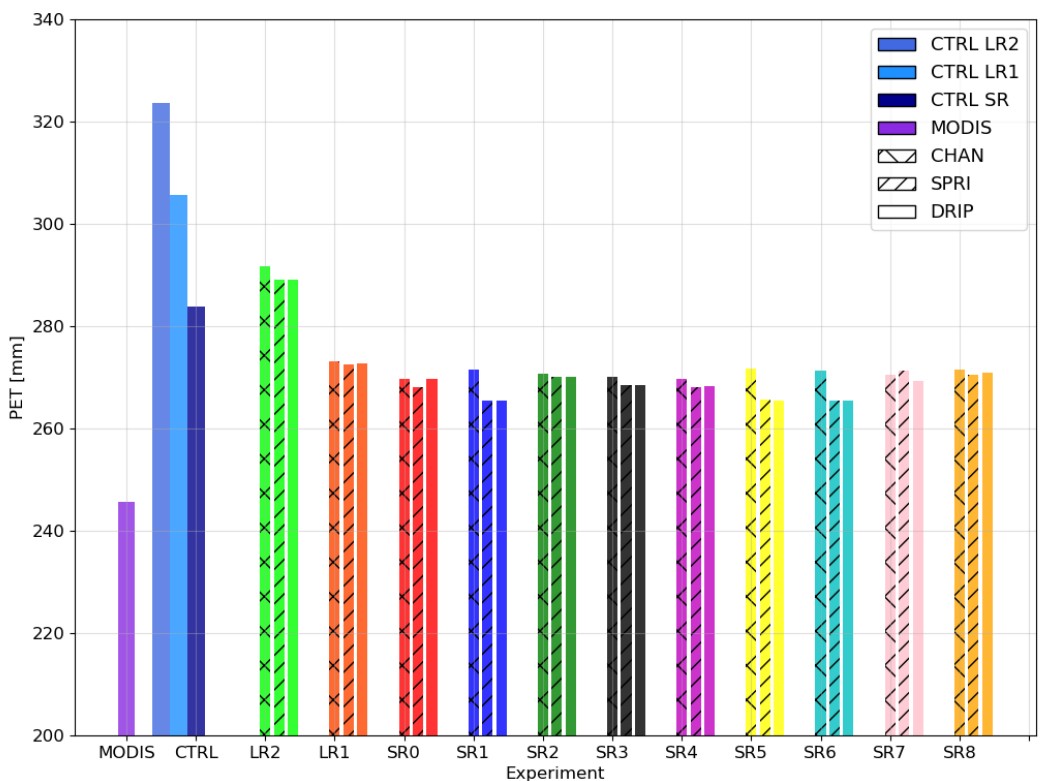

**Figure 8.** Monthly potential evapotranspiration accumulated for the irrigated area of the Po Valley (Fig.3)

with time scales longer than few days, such as soil moisture, is longer. Nevertheless, when the irrigation parameterization is
activated, the PET values are improved in all the experiments. In particular, the differences in SR and LR control run PET are
not observed anymore in the irrigated case of LR1 and SR0. Moreover, the potential evapotranspiration does not seems to be
affected significantly by the start, length and frequency of irrigation (SR experiments). There is some similarity between the
same starting time, especially considering differences in PET between the schemes when irrigation starts at 17 UTC (case 1, 5
and 6). However, such differences are very small compared to the quantities in play. Focusing on the frequency of the irrigation
with the coarse domain, it is clear that it does not have an effect on the potential evapotranspiration. In fact, the case 7 and 8,
respectively with frequencies of three and seven days, are similar to the cases SR0 and LR1.

There is no significant difference in the accumulated potential evaporation depending on the scheme used, only that the channel
shows slightly higher values. Nevertheless, all irrigation schemes improve the accumulated potential evapotranspiration.



## 5 Results and Discussion

### 5.1 Spatial Influence of Irrigation on Soil Moisture

Irrigation is applied to increase the water available to the plants, therefore in modeling terms has to influence the soil moisture in the simulation. Therefore, the spatial soil moisture changes induced by the three parameterizations are presented and discussed. Since the irrigation perturbation is applied regularly every day, and the temporal soil moisture scales usually are bigger than that, we expect that some memory is retained. To assess it quantitatively, spatial aware differences are used. This method allows having both temporal and spatial averages of the differences by averaging in correspondent dimension, without losing the spatial correlation of the introduced perturbation.

Firstly, we compare the soil moisture spatial differences of the long run, LR2, in the last simulated time-step between the irrigated parameterization run and the control one (Fig. 9). All methods show a similar increase in soil moisture with respect

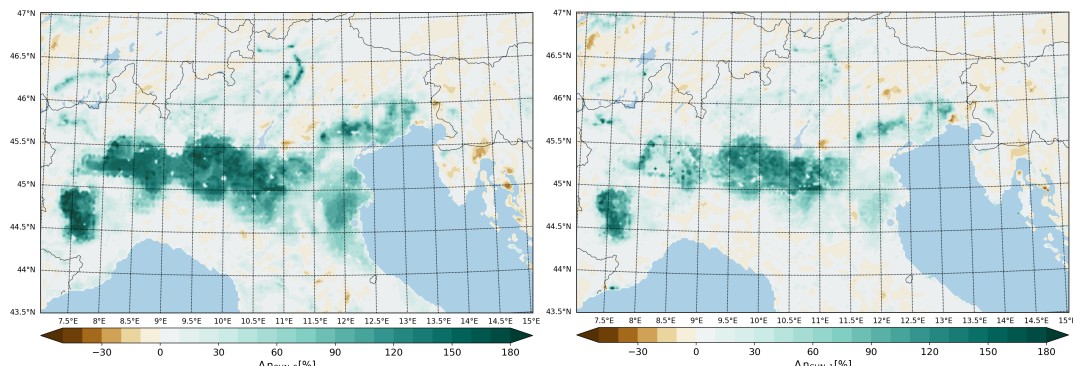

**Figure 9.** Last model timestep percentage changes of the irrigated run (Opt.1, CHAN) with respect to the control (D02) for both the first soil level (left) and the second one (right).

to the control run and a spatial pattern that is clearly related to the irrigation field of Fig. 3. For agricultural purposes, all soil layers are important since the water needs to reach the roots. Therefore, in assessing the spatial impact of soil moisture ($\eta$) both the first and second level are shown, which are respectively 10 and 30 cm thick. In irrigated agriculture, the root zone tends to be more shallow than in the non-irrigated one due the lack of competition between ground and surface water sources (Lv et al., 2010). Therefore, the first two layers are enough to capture the real root zone. As it is possible to observe from Fig. 9 both levels show an increase in soil moisture that is over $110\%$ for the first layer (on the left) and between $40 - 90\%$ for the second one. The different increase rate in soil moisture between the layers is caused by the different time scales in infiltration and loss by evaporation and/or runoff.

Since the main changes in soil moisture are located within the irrigated zone, most of the time series will be done for a spatial average over that area alone.



## 5.2 Scale Dependency of Irrigation Parameterization

Due to the high number of sensitivity combinations, the coarse domain is a more efficient way to run all the possible tests in terms of both computational costs and output storage. However, to use the coarse domain to run the sensitivity test, the main variables must not vary much between different resolutions if influenced by the same irrigation methods. Therefore, the three schemes are run for both resolutions in the standard configuration as TR1 and TR2 to test the resolution dependence.

Averaging over the irrigated area of both domains, two variables are considered: the two meter height temperature (T2) and
the soil moisture. While the use of soil moisture as a diagnostic has been previously discussed, the two meter temperature is a common parameter for atmospheric studies. Moreover, from the physical perspective, this variable is influenced by both the ground state and the atmosphere. Therefore, it is an ideal parameter to consider when investigating surface perturbations.

In this part, both the time series (left side of Fig. 10) of the variables and the differences between the different scales are shown. The latter are obtained as the differences between the field averages in different domains. So we subtract from the convection

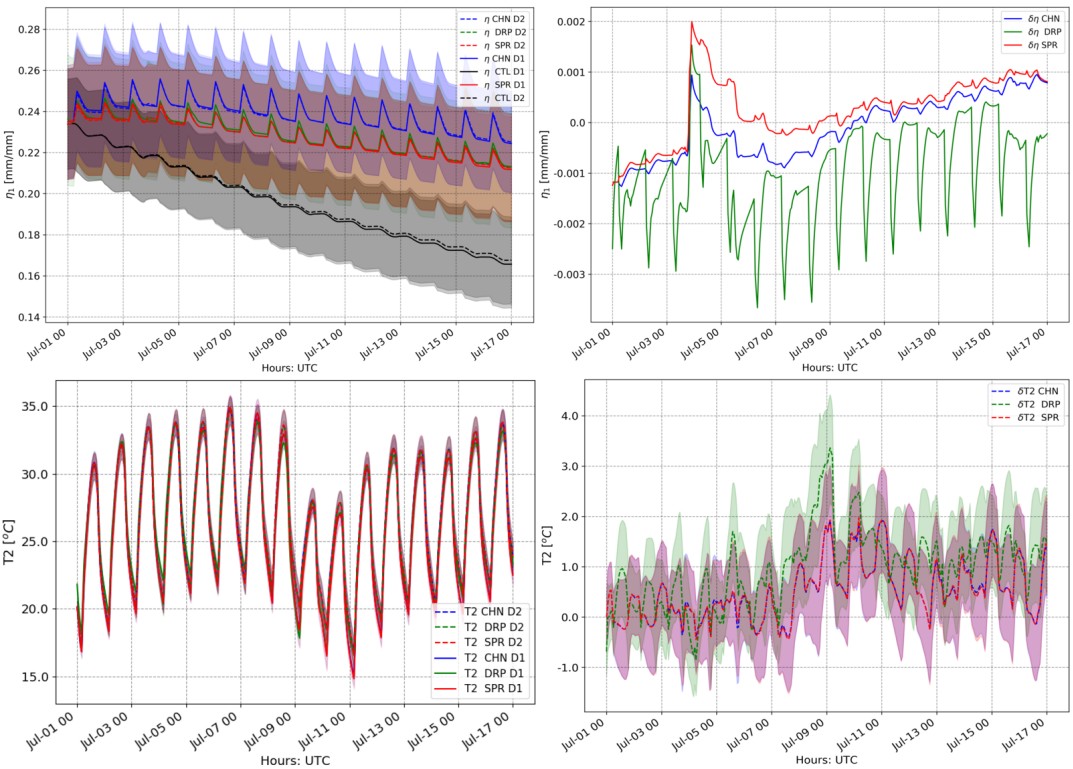

**Figure 10.** Time series of $\eta$(top panel) and T2 (bottom panel) for both domains and all three parameterizations averaged over the irrigated area, with the spatial standard deviation as shading. the differences in the time series is calculated and plotted respectively in the right side of the panel to highlight the differences between the resolutions. For the right panel, the shading represents the standard deviation of the differences.

permitting domain (D02) the value obtained in the convection parameterized one (D01). The results obtained are shown in the





right column of Fig. 10. The control simulation for TR1 and TR2 is added for the soil moisture field as well. When irrigation is activated, the soil does not dry as fast as in the control. The left side of the panel in Fig.10 shows that both variables have a similar behavior, within the spatial variability, in the two different scales. Since soil moisture is strongly affected by irrigation, which is not a spatially uniform field, the high standard deviation values are expected. This is the reason why the top right

figure of Fig. 10 does not show the standard deviation of either the high resolution domain nor the coarse one. In fact, the differences in soil moisture between the resolution is at most 1% of its value, and the spatial standard deviation is around $8\%$. Therefore, differences in soil moisture due to resolution can be considered negligible. Regarding the temperature, it is possible to observe that the differences are mostly in the second part of the simulation, i.e. after the $9^{th}$ of July when there was a small frontal passing. Most of the differences between the resolutions happen during the nighttime, while daytime is less affected.

One reason could be the cumulus scheme activation and its influence on the atmospheric state and dynamics. Nevertheless, the average behavior of the temperature fields are coherent with each other and with the different resolutions. Therefore it is acceptable to use the coarse resolution domain to understand the sensitivity of the parameterizations' assumptions, which is going to be shown in the next section.

### 5.3 Sensitivity

This part of the work discuss the sensitivity of the results to some of the parameterizations' assumptions, such as the irrigation start time and length, as well as the frequency of the irrigation. This part of the work investigates only the sensitivity run (SR) experimental settings (Table 2). Therefore, the SR nomenclature is dropped for now, so the parameterization and the case can be easily highlighted.

#### 5.3.1 Differences from the Control

First of all, the field time series of all the tests are shown in Fig.11 as differences from the control run (which is the non irrigated one) for both T2 and $\eta$. All simulations increase the soil moisture content with respect to the control run. In particular, left side of Fig. 11 highlights how the major differences between the single tests is driven by the scheme type more than the timing. Clearly then, the channel method (blue lines) is the one that shows the biggest increase in soil moisture, while both SPRINKLER and DRIP do not differ greatly from each other. Therefore, regarding irrigation efficiency, the atmospheric

evaporation in the SPRINKLER scheme is negligible if compared to the effect of the leaves and canopy interception, but the canopy interception is important in reducing the efficiency meaning that re-evaporation from the canopy provides a noticeable loss.

The effect on two meter temperature by the different assumptions is less evident than for soil moisture. In fact, most of the $\Delta$ time series in right side of Fig. 11 shows a decreasing in the mean daytime temperature up to $-1.5$ K. It is to notice how

the nighttime temperature of the channel parameterization are increased up to 0.7 K, while the other two are only up to 0.3 K. The time-series of the daily T2 minimum and maximum, for all run are considered in Fig. 12. Firstly, the maximum daily T2 is the quantity that more clearly is impacted by irrigation. For this quantity, all the SR irrigated tests behave very similarly by decreasing it with respect to the control run. The impact of the reduction of the maximum temperature is reduced outside the



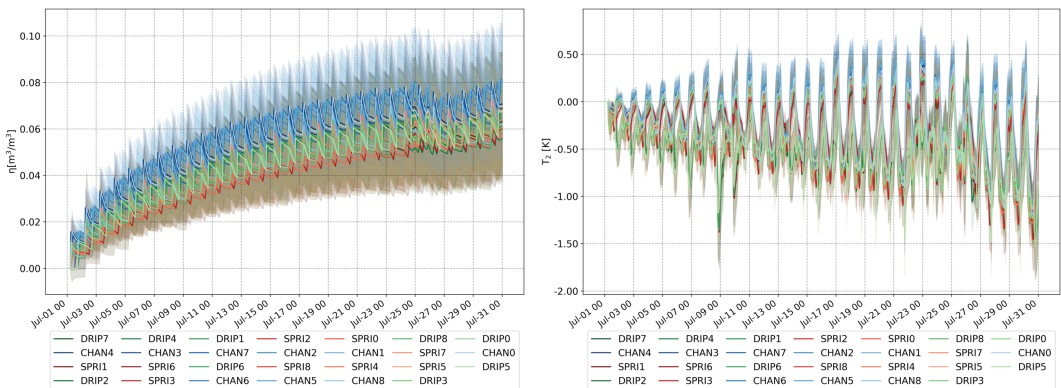

**Figure 11.** Time series of the differences from the control of all nine sensitivity tests averaged over the irrigated area. Soil moisture on the left and 2-meter temperature on the right. Blue colors shows the CHAN (Opt.1), green the DRIP (Opt.2) and red the SPRI (Opt.3).

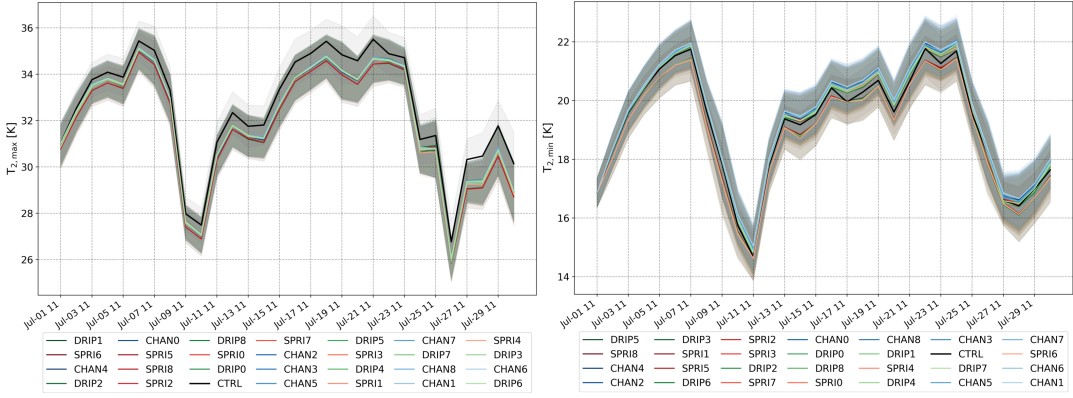

**Figure 12.** Time series of the daily maximum (left) and minimum (right) T2 averaged over the irrigated area for all sensitivity tests and the control. Blue colors shows the CHAN (Opt.1), green the DRIP (Opt.2) and red the SPRI (Opt.3).

main heat wave period, namely during the frontal passing of July 9 and 25-27. On the other hand, the minimum temperature
seems less impacted by irrigation itself as well as it's timing. All the SR behave similarly with the control, within the spatial standard deviation. Despite the fact that most of the irrigated SR simulations have higher minimum temperature than the control, the values are within the standard deviation. A probable reason for the warmer night temperature is the higher soil moisture. The higher $\eta$ gives the soil a larger thermal inertia since it increase the heat capacity.

### 5.3.2 Daily Cycle

The differences are taken with respect to the "standard" run (run 0) and not the control to analyze more in depth the effect of the parameterizations assumptions. This allows to isolate the effect of the specific assumption on the soil and atmospheric variables. Since the effects in most of the tests have a daily frequency, the average daily cycle is discussed. To understand the



temporal variability of the perturbation on the daily basis, with the diurnal mean cycle, also the single days are shown. In this part, in addition to the previously presented T2 and $\eta$, also the heat fluxes (SH) and the upward moisture fluxes, as well as the

soil temperature, are included.

Fig.13 (upper panel) shows that soil moisture differences from the standard are influenced by the timing and length of irrigation only regarding the peak location. In Fig. 13 is shown also the time series of the three standard run as well as the control as a reference. Firstly, it is possible to observe how the standard irrigated runs prevent the first layer of the soil to dry out, as it

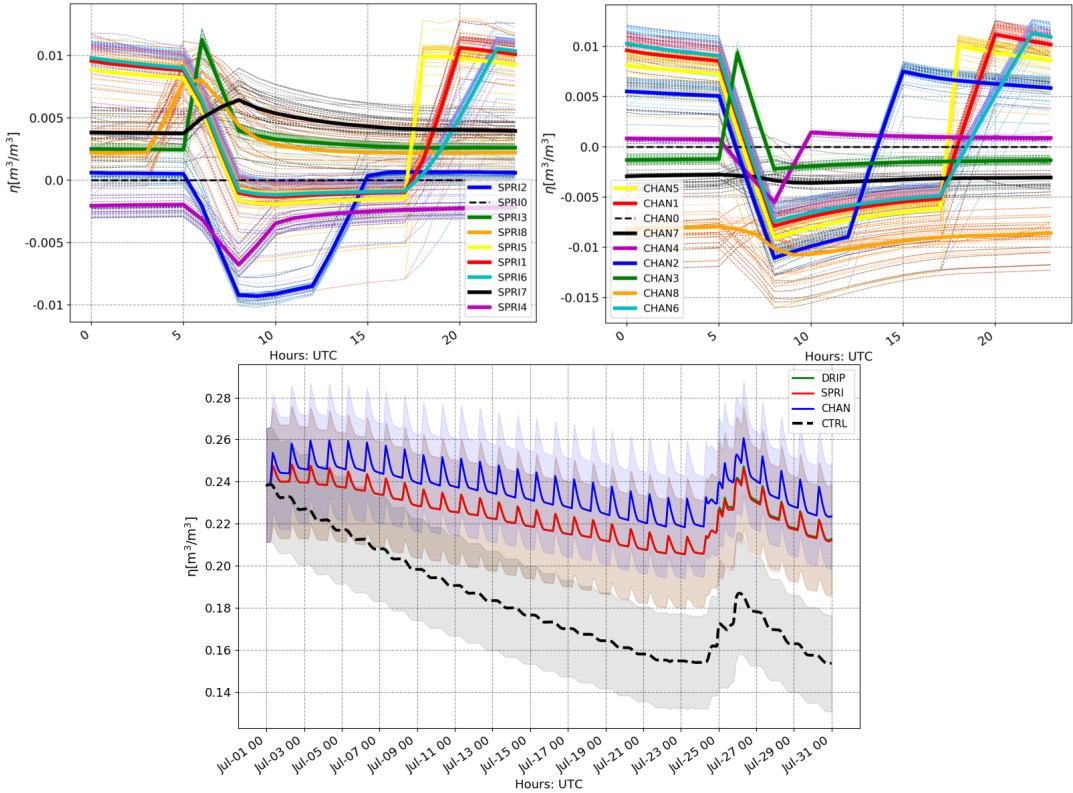

**Figure 13.** Upper panel: daily cycle of the mean difference between the test and the standard (run 0) in solid line, the single day differences are the light-dashed lines. Only two out of the three parameterizations are shown since there is no added information. Lower panel: time series of the three standard run and the control.

happens in the control. While the control run decreases from 0.24 to less than 0.16, the irrigated runs soil moisture always keep

over 0.22 (on average). Also, it is to notice that the channel parameterization soil moisture is always higher than both SPRI and DRIP run.

Despite having different baseline values for the standard simulation, it is possible to see that the maximum in the differences from the standard (Fig. 13 upper panel) accounts only for up to 4% of the total value. Moreover, the maxima in the differences are located accordingly to the different starting time and length. Also, there is no long term differences or multi-day trend since

the mean daily cycle agrees with the single diurnal cycles. On the other hand, the multi-day not in-phase irrigation is expected



to have a slightly different behavior on the daily bases. Despite the fact that all grid points will be irrigated within the selected period (three days for number 7, and seven days for number 8), the percentage of irrigated land within the points is different. This will lead to having some single days with different diurnal cycle with respect to others, and it will reflect as a larger spread observed in the daily cycles (e.g. Fig.13). Even though SR7 and SR8 have the same configuration of SR0, there is a change

in the diurnal cycle. However, it is to be noticed that such differences account only for less than 3% of the total soil moisture amount. Therefore, a multi-day frequency for irrigation does not seems to affect the longer term soil moisture trends.

Given the expected anti-correlation between soil moisture and temperature, the opposite behavior seen in the differences of T2 can be explained. However, when concerned with the larger scale impact, T2 could be used as an indicator of the atmospheric perturbation state. Fig. 14 shows the effect of timing and length on the SPRINKLER and the DRIP parameterization T2. Both

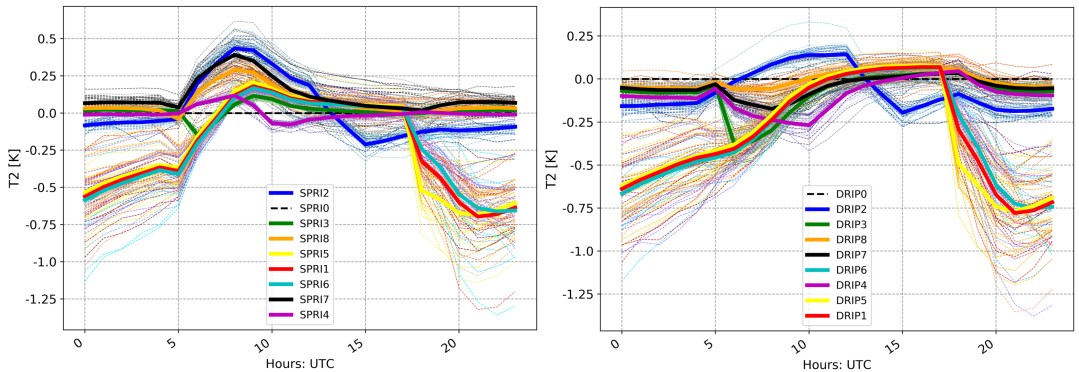

**Figure 14.** Same as Fig. 13 upper panel, for the two meter height temperature differences and a different set of parameterizations, for the SPRINKLER (Opt. 3) on the left and DRIP (Opt.2) on the right.

parameterizations show a higher day-to-day variability with respect to soil moisture, most likely due to the atmospheric state and dynamic influences. The bigger T2 differences in the DRIP scheme are mainly in the nighttime with the late-afternoon irrigation starting time (Fig. 14 right). On the other hand, the starting time at 12 UTC influences the daytime temperature more strongly in the sprinkler case than the DRIP one. This behavior is expected since the sprinkler directly affects the atmospheric state via the microphysics evaporation process that would be larger in the daytime. Also, the difference between daily frequency

in irrigation and the non in-phase run are almost negligible with the DRIP (Fig. 14 right) and slightly affected with the SPRI (Fig.14 left). A similar behavior to the DRIP is observed in the channel parameterization, which is not shown here, but the channel is warmer than the other two at night. While such differences in the night temperatures seems relevant for the local climate, it is to consider them in the context of Fig. 12 (right). In fact, while a nocturnal cooling of up to 1 K with respect to the standard run seems relevant, these magnitudes are within the spatial variability of this quantity.

When considering a change in the soil state, this will affect the energy flux partition at the surface. This part analyses only the DRIP, since the behavior of the differences with respect to the standard run are similar also for the other parameterizations. As for the soil moisture, also the fluxes differences are strongly affected by both timing and length, but only at the diurnal scale. In fact, there is no longer term trend underlying in the differences plot. The fluxes differences, Fig. 15 (upper panel), shows



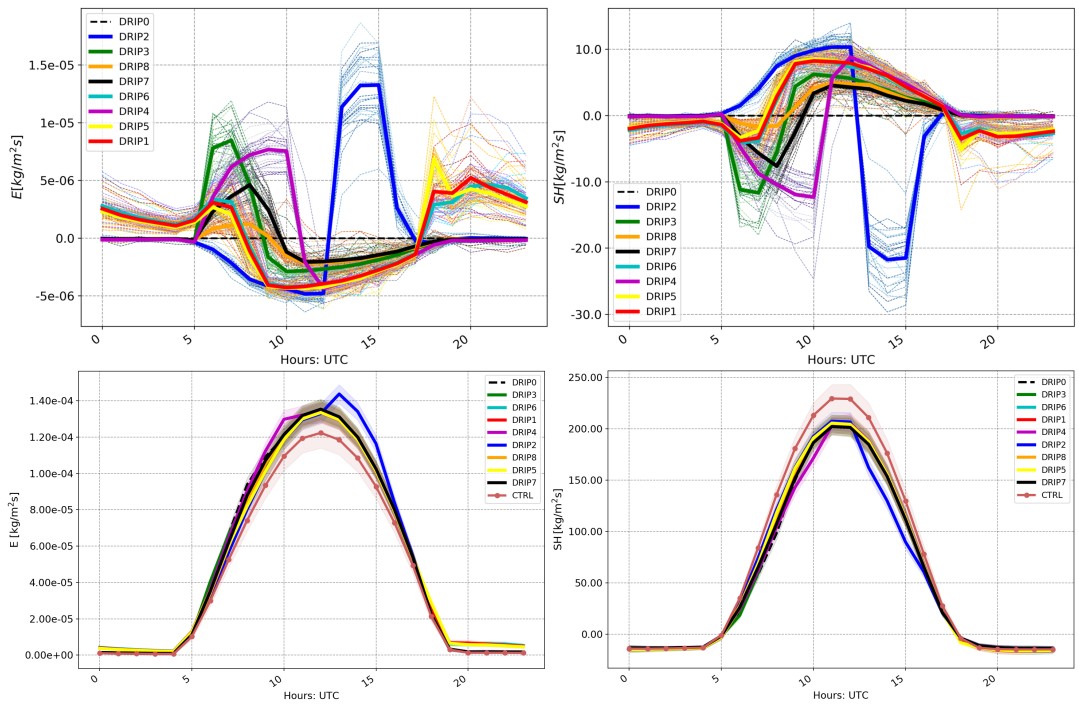

**Figure 15.** Same as Fig. 13; upward moisture and sensible flux for DRIP (Opt. 2), respectively on the left and right side, for the differences with respect to the standard run (upper panel) and for the absolute values of diurnal cycle monthly average (lower panel) including the control.

that the timing of irrigation impacts the fluxes mostly during the time when the parameterization is active. In particular, it is to
observe that the tests that differ most from the others are the ones where irrigation happens near the middle of the day (case number 4 and 2). Despite this, the differences from the other simulations they account for change of 12% ($E$) and 20% ($SH$) at most (Fig.15).

In the case of the afternoon irrigation (case 5,6 and 1), an increase in evaporation during most of the nighttime and a decrease during the daytime are observed. However, such decreases are usually small compared to the integrated increase. The described
behavior is reflected during the daytime, with opposite sign, in the sensible heat flux differences (Fig. 15 upper panel right). During the nighttime, on the other hand there are no significant differences between the tests, as can be observed comparing the different tests in the lower part of Fig. 15.

## 5.4 Irrigation efficiency

The methods described consider different processes regarding their modelled efficiency, after the water leaves the application
point. Therefore, it is possible to calculated the efficiency of the irrigation with the experiment performed.

There are different methods to estimate irrigation efficiency, but as irrigation is used to increase soil moisture the estimate





is done on this field. To assess the impact of the canopy evaporation (CANW), the percentage change in soil moisture field obtained with Opt. 2 (DRIP) with respect to the Opt. 1 (CHAN) is calculated. The use of, respectively, Opt. 3 (SPRI) and Opt. 2 (DRIP) allows to account for the impact of microphysics processes, as droplet evaporation and wind drift (EVDR).

This approach is valid under the assumption that irrigation is the only component of the model that affect the soil moisture field. While it might not be true for all the time-steps, it can gives a more accurate interpretation if it is applied only during the irrigated hours. The results obtained averaging over the irrigated area (Fig. 3) and for the whole July month, are shown in Tab. 4 below. The experiments show that the canopy/leaf interception effect is greater than the droplets evaporation and/or drift

**Table 4.** Efficiency expressed as soil moisture percentage change, for the whole irrigated domain in July, due to: EVDR, the impact of microphysics process (e.g. droplet evaporation), CANW the impact of canopy and leaves interception.

| Simulation experiment | EVDR [%] | CANW [%] |
|---|---|---|
| LR1 | -0.02 | -3.32 |
| LR2 | -0.09 | -3.53 |
| SR0 | -0.04 | -4.20 |
| SR1 | -0.02 | -3.45 |
| SR2 | -0.01 | -4.79 |
| SR3 | -0.06 | -3.42 |
| SR4 | -0.01 | -4.83 |
| SR5 | -0.06 | -3.52 |
| SR6 | -0.01 | -3.47 |
| SR7 | -0.02 | -2.11 |
| SR8 | -0.01 | -1.92 |

for all the experiments. A decrease around 4% of the soil moisture is obtained if the canopy effect is considered. The average

microphysic contribution, causes a decrease in soil moisture below the 0.1% for all the experiments. However, it is to notice that the EVDR value is higher in the LR2 test, the convection-permitting long simulation. This highlights a possible stronger modification and coupling of the local conditions at smaller scales, which is investigated in a separate study. As the SR7 and SR8 represent the non-daily frequency, the averaging process includes points which are not irrigated. This is reflected by the lower impact of the CANW component, but not on the EVDR one, which might be caused by the very low values.

## 6 Conclusions


Agricultural land plays important economical and social roles, as well as in influencing the local climate and biosphere. However, the land management change that impacts the local climate most is irrigation, if present. Recent literature shows that irrigation mostly affects the near surface variables, creating the so-called irrigation cooling effect. This local cooling found by different studies is on average of 1 K but up to 8 K, depending on the parameterization as well as the region. This study found



similar cooling impact of 1 K (with 1.03 for the CHAN and 1.17 for SPRI and DRIP), for the July spatial monthly average
difference. However, the maximum in the temperature cooling for this region reaches 4 K of cooling impact. Such differences
with previous limited area studies can be caused by the parameterizations choices, as it is found to be one of the uncertainties
sources of the irrigation impacts on climate (Wei et al., 2013; Leng et al., 2017, e.g.). The second cause is the water amount
applied, as none of the studies actually account for a realistic value. Several studies discussed that the lack of a common pa-

rameterization, as well as the non-explicit treatment of the amount of water used, is the main cause of the uncertainties.

This paper aims to present three new parameterizations for irrigation within the WRF-ARW model with an explicit water
amount. These parameterizations define different surface irrigation techniques based on the evaporation processes that the wa-
ter undergoes after it leaves the delivering system. Option 1 (CHANNEL) and 2 (DRIP) apply irrigation water as precipitation
as an input to the land surface model. While the Option 2 allows the interception of the water by the leaves and the canopy,

Opt.1 does not. Option 3 (sprinkler) irrigation water is affected by the microphysics processes (such as droplet evaporation or
drift), and it is introduced as rain mixing ratio in the lowest atmospheric mass-level of the model.

The current parameterizations are tested on one of the regions where global studies disagree on the signal of irrigation: the
Mediterranean area. In particular, the Po Valley in northern Italy is chosen due to the dense irrigation system and the vulner-
ability to heat waves. For this area, summer 2015 was a good test-bed season for the agricultural months. In fact, while both

summer months had positive temperature anomalies, their synoptic characterization was different. While, July 2015 was an
extreme month for high temperature anomalies, lack of precipitation and water stress, June 2015 was less extreme from the
hydrological cycle perspective.

In this study, for clarity, the options are called with names that recall existing techniques or the specific behavior. In fact, Opt.1
is defined channel method, as it resemble that historical technique. On the other hand, Opt. 3 is named sprinkler, as the water

is sprayed into the atmosphere and it uses the same assumptions of most of the irrigation studies within this discipline: the
droplets might undergo evaporation and/or drift (Uddin et al., 2010; Leng et al., 2017, e.g.). Option 2 is named DRIP, as the
water used for the irrigation is applied over the canopy only and then it can be intercepted and drip to the ground. The terms
here defined are not to be intended as an universal technique definition, but as a naming convention within this case study.

Three sets of experiment, with the same irrigation water amount of 5.7 mm/d, at the convection-permitting scale (3km here)

and/or parameterized (here, 15 km) are performed for this warm summer season or part of it. The 16-day test run (TR) is used
to assess the grid-scale dependency of irrigation's average field response. It is found that surface variables, such as 2-meter
temperature and soil moisture, do not show a different behavior depending on the model resolution. Therefore, a sensitivity
experiment (SR) to irrigation start time, length and frequency can be performed with the coarse setting. A long run (LR) of
3 months is used to validate the parameterizations against the surface temperature ground measurements from monitoring

weather stations. In fact, previous studies found that irrigation affects temperatures in both monthly averages of daily mean
and maximum. Therefore, the parameterizations are tested against these quantities, as well as the minimum temperature. On
average, the use of the irrigation schemes improve the model representation of these variables reducing the biases. In fact,
while for the control run the average bias is 0.75 $^o$C for the mean and 1.46$^o$C for the maximum temperature, the irrigation
ones are reduced respectively to $(-0.15 \pm 0.06)^o$C and $(-0.13 \pm 0.17)^o$C averaging the three methods. Then the July potential





evapotranspiration accumulated for the irrigated region of the Po Valley is evaluated for all nine sensitivity runs (SR0-8) and the long run (LR1 and LR2). All tests shows that the potential evapotranspiration is improved when the irrigation parameterization is used in the model.

Then the study addresses the sensitivity of the results to the parameterizations' human-decision assumptions: irrigation start time (in UTC), length and frequency (in days). The main impact of irrigation on soil moisture and 2-meter temperature with

respect to the control is due to the parameterization choice itself, rather then the timing. The channel method was slightly more efficient in terms of increasing the soil moisture long-term than the other two methods, but the similarity of the other two showed that the canopy interception was more important in reducing efficiency than the evaporation from the sprinkler process. Daily cycles of several atmospheric and surface variables are calculated as differences with respect to a standard configuration, namely when there is daily irrigation starting at 5 UTC and going for 3 hours. No significant impacts are found considering

all parameterizations for soil moisture, sensible heat and upward moisture fluxes. Assessing the impact of the T2 is more complicated, as the differences observed in the daily cycle are comparable to the ones that are obtained with different resolutions. While irrigation timing seems to affect the diurnal temperature cycle at the convection parameterized scale, other atmospheric variables (e.g. precipitation) do not seem to be affected. However, the precipitation effects are going to be presented in a different study.

The usage of an irrigation parameterization for this area improves the model representation. Moreover, on average, the atmosphere and soil variables are not very sensitive to the parameterizations' assumptions for realistic irrigation timing and length. Therefore, the use of the standard configuration alone for the high-resolution long run is acceptably representative.

Further analysis on assessing the physical and dynamical impact of the irrigation on the atmosphere is addressed in two follow-up works.

*Code and data availability.* Data and code are available at http://dx.doi.org/10.17632/t3b6rtccj9.1, cite this dataset as: Valmassoi, Arianna (2019), " Development of three new surface irrigation parameterizations in the WRF-ARW model: evaluation for the Po Valley (Italy) case study", Mendeley Data, v2.

## Appendix A: Surface Weather Stations Monthly Results

The values obtained for each station used in the validation section are written in Table A1. For clarity, the stations are identified

with an unique number (from 0 to 43) and their geographical coordinates, and not by their name. The temperatures value refers to the value obtained from the gridded model output. The bias is obtained subtracting to the simulation value the observation from the station.





**Table A1.** Monthly averaged mean, maximum and minimum values obtained for each station in the Fig. 5-7.

| Station number | Lon [E] | Lat [N] | Control T2 | bias | $T2_{max}$ | bias | $T2_{min}$ | bias | CHANNEL T2 | bias | $T2_{max}$ | bias | $T2_{min}$ | bias | SPRINKLER T2 | bias | $T2_{max}$ | bias | $T2_{min}$ | bias | DRIP T2 | bias | $T2_{max}$ | bias | $T2_{min}$ | bias |
|---|---|---|---|---|---|---|---|---|---|---|---|---|---|---|---|---|---|---|---|---|---|---|---|---|---|---|
| 0 | 11.126 | 44.826 | 27.38 | -0.61 | 35.30 | 0.94 | 18.35 | -2.73 | 26.95 | -1.04 | 34.01 | -0.35 | 18.75 | -2.32 | 26.85 | -1.14 | 34.45 | 0.09 | 18.38 | -2.70 | 26.77 | -1.22 | 33.98 | -0.38 | 18.51 | -2.56 |
| 1 | 11.016 | 44.886 | 27.63 | -0.04 | 35.27 | 0.98 | 18.75 | -1.45 | 26.96 | -0.71 | 33.76 | -0.52 | 19.04 | -1.15 | 26.87 | -0.80 | 34.19 | -0.09 | 18.76 | -1.44 | 26.78 | -0.89 | 33.73 | -0.56 | 18.87 | -1.32 |
| 2 | 10.147 | 44.743 | 28.19 | 0.40 | 34.92 | 1.56 | 19.96 | -1.71 | 27.77 | -0.03 | 33.85 | 0.49 | 20.47 | -1.20 | 27.54 | -0.25 | 34.20 | 0.84 | 20.25 | -1.43 | 27.62 | -0.17 | 33.83 | 0.46 | 20.29 | -1.38 |
| 3 | 10.259 | 44.952 | 28.27 | 0.95 | 35.75 | 2.83 | 18.94 | -1.84 | 27.36 | 0.04 | 33.75 | 0.84 | 19.70 | -1.09 | 27.18 | -0.14 | 34.13 | 1.21 | 19.41 | -1.38 | 27.18 | -0.14 | 33.74 | 0.82 | 19.46 | -1.32 |
| 4 | 10.350 | 44.944 | 28.27 | 1.97 | 35.72 | 3.32 | 18.98 | -0.50 | 27.43 | 1.12 | 33.88 | 1.48 | 19.58 | 0.10 | 27.27 | 0.96 | 34.26 | 1.86 | 19.18 | -0.30 | 27.27 | 0.97 | 33.91 | 1.51 | 19.30 | -0.18 |
| 5 | 10.773 | 44.743 | 27.57 | 0.02 | 35.45 | 1.28 | 18.33 | -1.65 | 26.98 | -0.57 | 33.94 | -0.23 | 18.92 | -1.05 | 26.99 | -0.56 | 34.51 | 0.34 | 18.68 | -1.29 | 26.91 | -0.64 | 34.02 | -0.15 | 18.72 | -1.25 |
| 6 | 10.381 | 44.885 | 28.05 | 0.64 | 35.76 | 1.99 | 18.60 | -1.91 | 27.18 | -0.24 | 33.87 | 0.11 | 19.20 | -1.31 | 26.97 | -0.44 | 34.28 | 0.52 | 18.79 | -1.72 | 27.03 | -0.39 | 33.94 | 0.18 | 18.94 | -1.57 |
| 7 | 10.971 | 44.778 | 27.49 | 0.52 | 35.38 | 2.16 | 18.16 | -1.64 | 27.14 | 0.17 | 34.17 | 0.95 | 18.75 | -1.05 | 27.13 | 0.17 | 34.73 | 1.51 | 18.44 | -1.36 | 27.02 | 0.05 | 34.20 | 0.98 | 18.54 | -1.26 |
| 8 | 11.512 | 44.886 | 27.27 | 0.03 | 34.88 | 1.19 | 18.79 | -1.47 | 26.86 | -0.38 | 33.67 | -0.01 | 19.41 | -0.84 | 26.75 | -0.49 | 34.10 | 0.41 | 19.13 | -1.12 | 26.83 | -0.42 | 33.79 | 0.11 | 19.30 | -0.95 |
| 9 | 11.896 | 44.968 | 26.83 | -0.02 | 33.92 | 0.81 | 18.93 | -1.55 | 26.55 | -0.30 | 33.08 | -0.03 | 19.35 | -1.13 | 26.47 | -0.38 | 33.50 | 0.39 | 19.13 | -1.35 | 26.56 | -0.30 | 33.17 | 0.06 | 19.28 | -1.20 |
| 10 | 11.126 | 44.826 | 27.38 | -0.61 | 35.30 | 0.94 | 18.35 | -2.73 | 26.95 | -1.04 | 34.01 | -0.35 | 18.75 | -2.32 | 26.85 | -1.14 | 34.45 | 0.09 | 18.38 | -2.70 | 26.77 | -1.22 | 33.98 | -0.38 | 18.51 | -2.56 |
| 11 | 10.206 | 44.703 | 27.67 | 0.32 | 33.71 | 1.57 | 20.55 | -1.54 | 27.29 | -0.05 | 32.75 | 0.60 | 20.91 | -1.18 | 27.20 | -0.15 | 33.18 | 1.03 | 20.90 | -1.19 | 27.14 | -0.21 | 32.68 | 0.53 | 20.85 | -1.24 |
| 12 | 11.483 | 44.749 | 26.86 | -0.15 | 35.09 | 0.88 | 17.50 | -1.85 | 26.26 | -0.75 | 33.68 | -0.53 | 17.80 | -1.54 | 26.12 | -0.89 | 34.17 | -0.05 | 17.44 | -1.90 | 26.14 | -0.86 | 33.77 | -0.44 | 17.54 | -1.80 |
| 13 | 10.773 | 44.743 | 27.57 | 0.02 | 35.45 | 1.28 | 18.33 | -1.65 | 26.98 | -0.57 | 33.94 | -0.23 | 18.92 | -1.05 | 26.99 | -0.56 | 34.51 | 0.34 | 18.68 | -1.29 | 26.91 | -0.64 | 34.02 | -0.15 | 18.72 | -1.25 |
| 14 | 11.337 | 44.922 | 27.44 | -0.23 | 34.99 | 1.04 | 18.99 | -1.98 | 26.91 | -0.76 | 33.70 | -0.26 | 19.35 | -1.63 | 26.82 | -0.85 | 34.10 | 0.14 | 19.09 | -1.88 | 26.80 | -0.87 | 33.72 | -0.24 | 19.22 | -1.76 |
| 15 | 9.590 | 45.041 | 28.79 | 1.58 | 35.66 | 2.99 | 20.63 | 0.02 | 27.80 | 0.58 | 33.67 | 0.99 | 20.76 | 0.15 | 27.65 | 0.44 | 34.00 | 1.32 | 20.63 | 0.03 | 27.69 | 0.48 | 33.65 | 0.97 | 20.68 | 0.07 |
| 16 | 10.511 | 44.690 | 27.86 | 0.36 | 35.31 | 1.76 | 18.94 | -1.48 | 27.26 | -0.23 | 33.90 | 0.35 | 19.60 | -0.82 | 27.14 | -0.36 | 34.42 | 0.87 | 19.34 | -1.09 | 27.04 | -0.46 | 33.87 | 0.32 | 19.36 | -1.06 |
| 17 | 11.337 | 44.922 | 27.44 | -0.23 | 34.99 | 1.04 | 18.99 | -1.98 | 26.91 | -0.76 | 33.70 | -0.26 | 19.35 | -1.63 | 26.82 | -0.85 | 34.10 | 0.14 | 19.09 | -1.88 | 26.80 | -0.87 | 33.72 | -0.24 | 19.22 | -1.76 |
| 18 | 10.168 | 45.007 | 28.48 | 1.59 | 35.67 | 2.95 | 19.48 | -0.82 | 27.54 | 0.64 | 33.66 | 0.94 | 20.09 | -0.21 | 27.42 | 0.52 | 34.06 | 1.35 | 19.87 | -0.44 | 27.40 | 0.50 | 33.66 | 0.94 | 19.93 | -0.38 |
| 19 | 10.005 | 45.003 | 28.24 | 0.86 | 35.86 | 2.53 | 18.85 | -1.72 | 27.17 | -0.21 | 33.67 | 0.58 | 19.47 | -1.10 | 26.91 | -0.48 | 33.97 | 0.64 | 19.18 | -1.40 | 26.92 | -0.47 | 33.53 | 0.20 | 19.20 | -1.38 |
| 20 | 10.909 | 44.551 | 27.93 | 0.34 | 35.14 | 1.22 | 19.11 | -1.10 | 27.68 | 0.09 | 34.14 | 0.22 | 20.12 | -0.10 | 27.62 | 0.03 | 34.68 | 0.76 | 19.91 | -0.30 | 27.55 | -0.04 | 34.14 | 0.22 | 19.89 | -0.32 |
| 21 | 8.989 | 45.281 | 29.08 | 1.06 | 35.50 | 0.88 | 21.91 | 0.46 | 27.87 | -0.15 | 33.46 | -1.16 | 21.73 | 0.28 | 27.71 | -0.31 | 33.63 | -0.99 | 21.61 | 0.15 | 27.71 | -0.31 | 33.25 | -1.37 | 21.62 | 0.17 |
| 22 | 10.664 | 45.263 | 28.39 | 0.23 | 34.78 | 0.34 | 20.10 | -1.07 | 27.41 | -0.75 | 32.93 | -1.51 | 20.62 | -0.55 | 27.28 | -0.88 | 33.28 | -1.16 | 20.46 | -0.72 | 27.26 | -0.89 | 32.84 | -1.60 | 20.53 | -0.65 |
| 23 | 10.684 | 45.412 | 28.49 | 0.30 | 34.35 | 1.11 | 21.16 | -0.48 | 27.68 | -0.50 | 32.67 | -0.58 | 21.52 | -0.12 | 27.48 | -0.71 | 32.91 | -0.34 | 21.24 | -0.39 | 27.51 | -0.67 | 32.57 | -0.68 | 21.42 | -0.21 |
| 24 | 11.290 | 45.015 | 27.69 | 0.01 | 34.83 | 2.37 | 19.64 | -2.77 | 27.12 | -0.56 | 33.52 | 1.06 | 19.78 | -2.64 | 27.08 | -0.60 | 33.93 | 1.47 | 19.61 | -2.80 | 27.05 | -0.62 | 33.55 | 1.09 | 19.73 | -2.68 |
| 25 | 9.147 | 45.179 | 29.94 | 2.21 | 35.93 | 3.26 | 23.68 | 1.17 | 29.19 | 1.45 | 34.29 | 1.61 | 23.50 | 1.00 | 29.04 | 1.31 | 34.49 | 1.82 | 23.50 | 1.00 | 29.10 | 1.36 | 34.18 | 1.50 | 23.51 | 1.00 |
| 26 | 9.521 | 45.443 | 28.49 | 0.47 | 34.75 | 0.23 | 20.60 | -0.39 | 27.06 | -0.96 | 32.34 | -2.19 | 20.66 | -0.52 | 26.84 | -1.18 | 32.46 | -2.07 | 20.53 | -0.45 | 26.88 | -1.14 | 32.19 | -2.34 | 20.60 | -0.38 |
| 27 | 9.612 | 45.621 | 30.21 | 2.14 | 34.46 | 1.84 | 25.29 | 2.42 | 29.41 | 1.35 | 32.80 | 0.18 | 25.25 | 2.37 | 29.30 | 1.23 | 32.94 | 0.32 | 25.47 | 2.60 | 29.29 | 1.23 | 32.66 | 0.04 | 25.25 | 2.38 |
| 28 | 10.195 | 45.121 | 28.74 | 1.39 | 35.32 | 0.81 | 20.54 | 0.70 | 27.48 | 0.13 | 33.25 | -1.27 | 20.74 | 0.90 | 27.28 | -0.06 | 33.49 | -1.03 | 20.50 | 0.67 | 27.32 | -0.02 | 33.15 | -1.36 | 20.55 | 0.72 |
| 29 | 9.135 | 45.324 | 28.92 | 1.36 | 35.19 | 2.09 | 21.64 | -0.34 | 27.72 | 0.16 | 33.10 | 0.00 | 21.48 | -0.50 | 27.51 | -0.05 | 33.26 | 0.16 | 21.34 | -0.64 | 27.58 | 0.02 | 32.93 | -0.17 | 21.40 | -0.58 |
| 30 | 10.059 | 45.163 | 28.89 | 0.65 | 35.35 | 0.45 | 20.69 | -0.27 | 27.55 | -0.69 | 33.22 | -1.42 | 20.89 | -0.06 | 27.36 | -0.87 | 33.48 | -1.41 | 20.63 | -0.35 | 27.46 | -0.78 | 33.17 | -1.73 | 20.78 | -0.17 |
| 31 | 10.798 | 45.157 | 30.07 | 0.77 | 35.40 | -0.41 | 24.11 | 1.12 | 29.50 | 0.20 | 34.03 | -1.78 | 24.31 | 1.32 | 29.43 | 0.14 | 34.33 | -1.48 | 24.27 | 1.28 | 29.42 | 0.12 | 33.97 | -1.84 | 24.29 | 1.31 |
| 32 | 9.891 | 45.398 | 29.01 | 1.23 | 34.92 | 1.99 | 21.55 | -0.43 | 27.59 | -0.19 | 32.65 | -0.28 | 21.45 | -0.54 | 27.36 | -0.42 | 32.80 | -0.13 | 21.28 | -0.71 | 27.35 | -0.43 | 32.47 | -0.46 | 21.25 | -0.74 |
| 33 | 9.964 | 45.255 | 29.06 | 1.47 | 35.14 | 0.91 | 21.62 | 1.15 | 27.71 | 0.12 | 33.00 | -1.22 | 21.39 | 0.92 | 27.48 | -0.11 | 33.16 | -1.06 | 21.16 | 0.69 | 27.52 | -0.08 | 32.85 | -1.38 | 21.20 | 0.73 |
| 34 | 9.354 | 45.472 | 28.72 | 0.46 | 34.61 | 0.95 | 21.32 | -1.51 | 27.68 | -0.59 | 32.65 | -1.00 | 21.52 | -1.57 | 27.48 | -0.78 | 32.81 | -0.85 | 21.29 | -1.54 | 27.48 | -0.78 | 32.48 | -1.18 | 21.13 | -1.70 |
| 35 | 10.768 | 44.964 | 27.88 | -0.50 | 35.26 | -0.14 | 18.90 | -1.80 | 26.82 | -1.56 | 33.37 | -2.02 | 19.26 | -1.44 | 26.73 | -1.65 | 33.77 | -1.62 | 19.04 | -1.66 | 26.72 | -1.67 | 33.42 | -1.98 | 19.17 | -1.53 |
| 36 | 10.887 | 45.188 | 28.63 | 1.18 | 34.90 | 0.64 | 20.96 | 1.25 | 27.74 | 0.28 | 33.30 | -0.96 | 21.20 | 1.50 | 27.67 | 0.21 | 33.65 | -0.61 | 21.07 | 1.36 | 27.64 | 0.18 | 33.25 | -1.01 | 21.13 | 1.43 |
| 37 | 9.105 | 45.432 | 30.56 | 2.72 | 35.40 | 2.31 | 25.45 | 2.56 | 29.45 | 2.56 | 34.20 | 1.11 | 25.45 | 2.56 | 29.84 | 2.01 | 34.01 | 0.92 | 25.51 | 2.62 | 29.90 | 2.06 | 33.71 | 0.63 | 25.45 | 2.56 |
| 38 | 9.380 | 45.260 | 28.66 | 0.31 | 35.16 | 0.66 | 21.01 | -1.13 | 27.38 | -0.96 | 32.98 | -1.51 | 20.85 | -1.29 | 27.12 | -1.23 | 33.09 | -1.41 | 20.71 | -1.43 | 27.22 | -1.12 | 32.82 | -1.68 | 20.78 | -1.36 |
| 39 | 9.276 | 45.233 | 29.00 | 1.00 | 35.41 | 2.40 | 21.62 | -0.78 | 27.65 | -0.35 | 33.18 | 0.17 | 21.32 | -1.08 | 27.41 | -0.59 | 33.31 | 0.30 | 21.14 | -1.26 | 27.52 | -0.48 | 33.03 | 0.02 | 21.24 | -1.16 |
| 40 | 8.880 | 45.341 | 30.18 | 3.06 | 35.72 | 3.10 | 24.31 | 3.23 | 29.34 | 2.22 | 34.00 | 1.38 | 24.20 | 3.12 | 29.20 | 2.07 | 34.18 | 1.56 | 24.14 | 3.07 | 29.23 | 2.11 | 33.84 | 1.22 | 24.20 | 3.12 |
| 41 | 9.692 | 45.715 | 29.60 | 2.09 | 33.40 | 0.54 | 25.42 | 3.50 | 29.02 | 1.52 | 32.17 | -0.70 | 25.30 | 3.39 | 28.94 | 1.43 | 32.29 | -0.57 | 25.32 | 3.41 | 28.95 | 1.44 | 32.06 | -0.81 | 25.21 | 3.30 |
| 42 | 9.487 | 45.186 | 28.16 | 0.12 | 35.30 | 2.25 | 19.76 | -2.32 | 26.87 | -1.16 | 32.84 | -0.21 | 19.97 | -2.10 | 26.59 | -1.44 | 32.96 | -0.09 | 19.69 | -2.39 | 26.66 | -1.37 | 32.65 | -0.40 | 19.77 | -2.31 |
| 43 | 9.822 | 45.784 | 27.58 | 1.71 | 31.26 | 0.21 | 23.61 | 2.88 | 27.02 | 1.14 | 30.16 | -0.90 | 23.26 | 2.53 | 26.95 | 1.07 | 30.32 | -0.74 | 23.35 | 2.62 | 26.97 | 1.09 | 30.08 | -0.98 | 23.19 | 2.46 |

*Author contributions.* AV and JD designed the methodology and the experiments, with the help of FP and SDS. AV developed the model code, performed the simulations and the analysis under the supervision of JD. JD provided the computational resources. AV prepared the manuscript under the supervision of JD and FP. JD, FP, and SDS reviewed the manuscript and provided the funding.

*Competing interests.* The authors declare that they have no conflict of interest.

*Acknowledgements.* The model simulations used for this study are available at http://dx.doi.org/10.17632/t3b6rtccj9.1. Weather stations' data and MODIS potential evapotranspiration data have to be obtained from the respective agencies, in agreement to their data policies. This work is carried out as part of the National Center for Atmospheric Research (NCAR) Advanced Study Graduate Visitor Program (ASP),




the iSCAPE (Improving Smart Control of Air Pollution in Europe) project (funded by the European Union's Horizon 2020 research and innovation programme H2020-SC5-04-2015 under the Grant Agreement No. 689954) and the OPERANDUM (OPEn-air laboRAtories for Nature baseD solUtions to Manage hydro-meteo risks) project (funded by the European Union's Horizon 2020 research and innovation programme under Grant Agreement No 776848). Cheyenne computational resources are provided by ASP through the Computational & Information System Lab (CISL) funded by the National Science Foundation (NSF).

The authors declare that there was no conflict of interest.





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
