# Peer review of "Evaluation of three new surface irrigation parameterizations in the WRF-ARW v3.8.1 model: the Po Valley (Italy) case study"

_Geoscientific Model Development, 2019_

## Referee Comment (RC1) · Anonymous Referee #1 · 2 Dec 2019

**MAJOR REMARKS**

The authors present a sensitivity study on simulating the effects of irrigation in the Po valley with the WRF model. They conducted a range of sensitivity simulations using three types of simplified irrigation parameterizations and different assumptions about the timing and frequency of the irrigation. In addition, they considered two different WRF resolutions over the domain of interest. The topic of the paper is interesting and the results of the study have the potential to increase our knowledge on the importance of irrigation for the regional climate of the Po valley and on the suitability of the three parameterizations for its usage regional climate modelling. Unfortunately, the paper

**suffers from several flaws that need to be removed before the paper may be published.**

Many details are provided with regard to the effect of irrigation and partially it seems that they are sold as new results. However, effects are neither new nor surprising as they can be expected from process knowledge and previous studies. For example, it is well known that irrigation increases soil moisture and latent heat flux, lowers sensible heat flux, surface and 2m temperatures and often reduces related model biases that originate from not representing irrigation in respective areas. Also, a reduced irrigation effect can be expected for the DRIP and Sprinkler irrigation due to the regarded canopy interception and related evaporation. Interesting and new are the differences in magnitude of these effects between the different irrigation types and the different assumptions. While the irrigation effects on surface variables is well generally known, the impact of the irrigation on the climate of the Po valley would be a rather intriguing information, but unfortunately such effects are moved to follow up studies. I found this somewhat puzzling, as many other studies that applied irrigation in climate modelling consider effects on surface and climate at once in the respective publication.

One major obstacle in reviewing and understanding the paper is the low quality of the English. Several times, I was not able to understand a sentence. Usually, I also provide some suggestions for text edits and sentence improvements, but the high amount of unclear sentences, wrong gramma and unusual sentence structures let me refrain from doing this. I strongly recommend a careful proof reading from a native English speaker.

In addition I am wondering why the study's results have been submitted to GMD. The study comprises the results of a set of sensitivity studies that have been conducted with an existing regional climate model using three types of simplified irrigation parameterizations. I would not consider the respective simplified equations as new model development. Moreover, even though the total amount of irrigation may be consistent with actual irrigation amounts, the water used for irrigation is taken out of nowhere, i.e. water conservation dos not play a role, which I would expect if something is introduced as a new model. Therefore, I believe that other journals such as HESS may be more
appropriate for the publication of the study.

Consequently, I suggest focusing on the new results of the study and reducing the details in describing known and expected effects. The paper would strongly benefit from including irrigation effects on climate variables, especially precipitation. In summary, I suggest making major revisions and resubmitting the paper to a more suitable Copernicus journal.

**Minor remarks**

In the introductory section, some links to previous studies on irrigation seem to be missing. As the present study is dealing with uncertainties related to the representation of irrigation in a climate model, it should be noted that there have been a few studies, which aim at the effects related to differences in the representation of irrigation within models as a key factor of uncertainty. In these studies, the focus was mainly on the extent of the irrigated areas or timing and mode of delivery, e.g. Sacks et al. (2009) and Yoshikawa et al. (2013). However, recently deVrese and Hagemann (2018) investigated uncertainties related to the representation of irrigation characteristics with respect to irrigation effectiveness and the timing of the delivery. Further, it is written on p3, line 60-61, that "Moreover, most importantly the scheme proposed do not account explicitly for irrigation water amount as an input." I do not agree with this statement as several climate model studies exists where the water for irrigation is withdrawn from existing reservoirs within the respective model framework (e.g., from reservoirs of the river routing scheme such as in Guimberteau et al. 2012, deVrese et al. 2018), or where the amount of irrigation is limited by using information from observed river runoff (Saeed et al. 2012). When interpreting the results of the present study, it should be kept in mind that the impact of irrigation on simulated climate may also differ between models or simulations despite identical assumptions about the irrigation characteristics (Tuinenburg et al. 2014; Krakauer et al. 2016).

References de Vrese, P., and S. Hagemann (2018) Uncertainties in modelling the cli-
mate impact of irrigation. Clim. Dyn., 51:2023, doi: 10.1007/s00382-017-3996-z. 108.

de Vrese, P., T. Stacke & S. Hagemann (2018) Exploring the biogeophysical limits of global food production under different climate change scenarios. Earth Syst. Dyn., 9: 393-412.

Guimberteau M, Laval K, Perrier A, Polcher J (2012) Global effect of irrigation and its impact on the onset of the Indian summer monsoon. Clim Dyn 39:1329–1348

Krakauer NY, Puma MJ, Cook BI, Gentine P, Nazarenko L (2016) Ocean–atmosphere interactions modulate irrigations climate impacts. Earth Syst Dyn 7(4):863–876

Sacks WJ, Cook BI, Buenning N, Levis S, Helkowski JH (2009) Effects of global irrigation on the near-surface climate. Clim Dyn 33(2–3):159–175

Saeed F, Hagemann S, Saeed S, Jacob D (2012) Influence of mid-latitude circulation on upper Indus basin precipitation: the explicit role of irrigation. Clim Dyn. doi:10.1007/s00382-012-1480-3

Tuinenburg O, Hutjes R, Stacke T, Wiltshire A, Lucas-Picher P (2014) Effects of irrigation in india on the atmospheric water budget. J Hydrometeorol 15(3):1028–1050

Yoshikawa S, Cho J, Yamada H, Hanasaki N, Khajuria A, Kanae S (2013) An assessment of global net irrigation water requirements from various water supply sources to sustain irrigation: rivers and reservoirs (1960–2000 and 2050). HESSD 10(1):1251–1288

p. 5 – eq. 1 and line 127-128, Table 2 I don't understand the definition of iADT1. Why a frequency is characterized with the difference symbol iAD? Why a frequency is expressed in number of days and not number per days? If irrigation is conducted once per weak, I would expect a frequency of 1/7 days, and not 7 days such as it is defined.

p. 5 – line 134-135 It is written: "Therefore, the water accumulated on the canopy is imposed to zero when irrigation is activated."
Does this mean that (as irrigation water is added to precipitation) also the actual precipitation is not intercepted? This would introduce an erroneous model change that makes the comparison to the control simulation invalid as effects on model results are not only caused by irrigation itself.

p. 7 – line 180-181 Why a non-reproducible option is given as default, and not the reproducible one? This makes a potential code debugging more difficult.

p. 7 -line 189 Do you need to pay regard to a buffer zone near the boundary in which spurious boundary effects may occur?

p. 8 – line 195 Information of NCEP is not relevant for this paper.  $\rightarrow$  Remove!

p. 8 – line 206-208 Redundant information in these two sentences.  $\rightarrow$  Merge and remove redundancy!

p. 12 – Fig.4 - caption Panel description is missing.

p. 13 - line 311 In Fig. 3, no biases are plotted. I assume you mean Fig. 5?

p. 14 – line 332-333 It is written: "presents a bigger increase in the negative biases in highly irrigated areas, and a lesser decrease of the positive ones in area with low percentage of irrigated land." This is unclearly written. Do you mean that negative bias get bigger (as it is written), or do you mean that you get more points with negative biases (what I infer from Fig. 5).

p. 16 – line 354-355 Why do you not use the coarsest scale of the model, i.e. the 15 km resolution?

p. 17 – Fig. 17 and related text The MODIS PET is an estimated product, and do not comprise real observations. What are the uncertainties of the MODIS PET data? I assume that there are larger uncertainties. Consequently, I suggest writing clearly that results change with respect to MODIS, and not that simulated PET becomes better. For example, in line 365, I suggest writing "PET values are closer to MODIS in all

GMDD
experiments" instead of "PET values are improved in all the experiments".

p. 18 - line 379 What are "spatial aware" differences? Please define thoroughly!

p. 18 – Fig. 9 - caption "...percentage changes " of what? p. 21 – Fig. 11 and 12 Figures are too busy. Showing 28 different curves, the curves are not distinguishable.

p. 22, 23, 24 – Fig. 13 (upper panels) and 14, 15 (upper panels) Figures are too busy and blurry with all these curves. The light-dashed lines do not provide any additive value and strongly distort the figures.

p. 25 – Table 4 and related text Why do you express the efficiency as a percentage change of soil moisture? In this way, the efficiency inherently depends on the absolute saturation/wetness of the soil, so that the efficiency cannot be adequately compared with studies from other regions, years or season. For example, if an irrigation method leads to 10 mm more water in the soil over a certain period, its efficiency will vary depending on the initial soil moisture, e.g. for soils with 20 mm or 30 mm initial content, this would yield 50% or 33% efficiency. However, I would expect the efficiencies are defined based on the total irrigation water amounts. For example, you may define an efficiency as the ration of the soil moisture change and the irrigated water. For example, if the latter is 40 mm, then for both soils mentioned above, the efficiency would be 25%.

In addition, please clearly define EVDR and CANW. I assume, CANW = DRIP-CHAN and EVDR=SPRINKLER-DRIP?!

p. 28 – Table A1 The character size is much too small.

GMDD

---

## Referee Comment (RC2) · Anonymous Referee #2 · 3 Dec 2019

The paper presents the development of three irrigation parametrization schemes in WRF and evaluates these schemes for the Po Valley (Italy). The paper fits well with the scope of GMD. Showing the irrigation cooling is not something new at all. But, showing that atmospheric and soil variables are not very sensitive to the parametrization assumption for irrigation timing and length is both interesting and questionable. Overall, the paper requires a significant revision to (a) bring the present study into the context of previous studies (i.e., address the novelty or difference of this study), (b) improve the figure quality and presentation quality (to reduce clutter and improve he text flow), and (c) discuss how the assumption and surface canopy may affect the results.

1. Introduction. The introduction, as it written, is a bit narrow and doesn't reflect the state of science in understating and/or modeling of irrigation effects. More references (or referring readers to other papers) should be added.

a) The introduction mentions little about the central pivot irrigation that is the main method for many parts of framing in United States. In addition, over the central Great plains, underground water is pumped for irrigation, and that water can be 10-20C lower than the surface water. Can the three irrigation schemes be applied to the central pivot irrigation from ground water?

b) The irrigation has important secondary effect on atmospheric dynamics, clouds, precipitation and infrared radiation (as water vapor is a greenhouse gas). The introduction lacks the summary of these secondary effects, or at least should refers the readers on these effect to previous work (ie., see the literature review by Aegerter et al., 2017 already cited in the manuscript). In addition, L45-48, page 2: Aegerter et al. 2017 actually find the surface cooling by irrigation can lead to regional subsidence and so decrease of cloud fraction, which is different from Qian et al. (2013).

c) There are already lots of irrigation schemes. A table summarizing these schemes and comparing/contrasting these schemes with the three schemes developed in this study should be made. Note, in reality, the soil moisture is never 100% in its field capacity for the whole growing season. In average, 50% is more to the norm. See Aegerter et al., 2017. This should be pointed out after the text in L55-60.

d) The last paragraph should also talk about the canopy effect as a result of irrigation, which is addressed in several past studies (such as Qian et al., Aegerter et al.). Without irrigation, there would not be canopy/crops, and the surface albedo would change. The schemes and experiments in this paper don't seek to address that, but this should be made clear and discuss the likely impacts (based on the past work).

2. Irrigation parameterization.

a. How does the development here differ from the paper by Lawston et al., 2015, J. Hydrometeor., 1135–1154?

b. Does the scheme consider the evaporation of water on the leaves (and so the cooling effect)? Does the temperature of irrigated water matter?

c. Irrigation mask field. The work here is similar as Aegerter et al., 2017 in which MODIS-based USDA irrigation database was used. However, it remains unclear how the fraction/percentage of irrigation in a model gridbox is factored into the Noah Land Surface Model in terms of surface properties for that gridbox as whole.

d. Does the crop types matter over the irrigated area? Aegerter et al. designate that as irrigated cropland and pasture for CLM. How the albedo, leaf area index or NDVI are specified for crops over the irrigated area in NOAH? Obviously, these are the parameters/questions that the present manuscript is not trying to address, but it is important to be clear about it.

3. Method

a) The method section only briefly mentioned that Noah LSM is used. It is unclear how the soil moisture responses to the rainfall/irrigation. Is the reference of Ek 2003 the most recent paper for Noah LSM? How surface energy budget is modeled in general terms? What surface type database is used? In Figure 2, there are 12 croplands. Are all these croplands irrigated? Does Noah treat these 12 croplands differently in terms of their albedo, leaf area index or NDVI?

b) L240-245. Where does 7mm/day come from? Should there be more irrigation in the early stage of growing season?

c) Control and sensitivity experiments didn't consider the canopy effect. Is this important? Note, if just irrigation (with no crop growing), should the surface be warmer or cooler? No irrigation and no crops should be the baseline experiment on top of which the irrigation effect can be fully studied (Aegerter et al. 2017).

4. Validation and results. The presentation here and the text flow are difficult for readers to comprehend.

a. Table 3. Should this monthly bias? The title says "the monthly values of mean T2". If so, why the minimum temperature (x̂bar) can be below 0 (the right most column) during the growing season?

b. Figure 5, 6, and 7 have significant repeating. The font size is too small; the labels are misleading – should it be labeled as \delta max T2 in figure 6, and \delta min T2 in Figure 7? The legend for the dot size is all shown in blue color, but the actual data dots show the red color as well. This is very confusing. Can these figures be summarized by showing area-averaged temperature as a function time, separated for upper left t part of stations and lower right of the stations? Some figures can be moved to the supporting materials.

c. Figure 8. Why the colors are different between legend and bar color?

d. Fig. 10 and Fig. 11 look similar. Again, the font size is too small for this reader to read. So is Figure 12.

e. Figure 13. Mean difference of what? Soil moisture? At which level? What are the differences between top left and top right panel? Font size is too small here again.

f. Figure 14. What is shown here is the difference with respect to the control? How about the difference with respect to the observed T (averaged over all stations)?

g. The irrigation efficiency does depend on the leaf area. In the early growing season, the crop height is low and leaves are small. The efficiency should be similar. With all the assumptions made, it is questionable if the parametrization schemes here have the fidelity to address the issue of irrigation efficiency. From an economic point of view, farmers use irrigation to grow crops, and so, the irrigation amount is unlikely uniform throughout the growing season (as assumed by the model here). Taller crops may need more water, and so, what is the point of evaluate irrigation efficiency if the specific crop

types are not considered? In addition, there are issues about the cost for each irrigation method. Overall, section 5.4 is cursory and is recommended to be removed from the text.

---

## Author Comment (AC1) · 31 Jan 2020

R1:
Major remarks, answers:

1. C: Many details are provided with regard to the effect of irrigation and partially it seems that they are sold as new results. However, effects are neither new nor surprising as they can be expected from process knowledge and previous studies.
   A: The physical explanation of the irrigation impact is not intended as new, rather a cross-check of the performance of the schemes with respect to previous global

studies. This part is important as older studies (Sacks et al, 2009 and Boucher et al. 2004) found that irrigation increases the surface temperatures in the Po Valley, and Thiery et al. 2017 found a decrease. The aim of this paper is to introduce and validate the irrigation parameterization, in the context of a regional model. The physical responses are discussed in two others paper due to length constraints (one can be found at: https://www.mdpi.com/2073-4433/11/1/72 while the other is accepted pending revisions). New aspects are also the comparison of the impacts of timing as well as the evaporative loss.

2. C: I am wondering why the study's results have been submitted to GMD. The study comprises the results of a set of sensitivity studies that have been conducted with an existing regional climate model using three types of simplified irrigation parameterizations. I would not consider the respective simplified equations as new model development.
A: The current work is submitted in GMD as it addresses three new parameterizations that were not previously included in WRF, to be released in the model such schemes have to be properly documented in journal publication. Moreover, there is no current irrigation parameterization in mesoscale models which is available for studies that constrains water used and allows timing as input. The parameterizations' limitations has to be contextualized with LAM models and their limitations in representing the water cycle. Also, past equations (which consider just soil moisture saturation,e.g.) are not suitable to represent irrigation processes at the regional scale, especially when going towards convection-permitting ones.

Minor remarks, answers:

1. C: However, recently deVrese and Hagemann (2018)investigated uncertainties related to the representation of irrigation characteristics with respect to irrigation effectiveness and the timing of the delivery.
A: We are going to add the new references of the previous studies, as it further

helps to prove the point of the need for a timing investigations: de Vrese et al, 2018 only investigates the differences between irrigating every model timestep (not realistic) and bi-weekly. The assumptions of previous global studies are likely correct at the investigated resolution of order-100 km, but can be improved at the regional scales and for mesoscale studies.

2. C: I do not agree with this statement as several climate model studies exists where the water for irrigation is withdrawn from existing reservoirs within the respective model framework (e.g., from reservoirs of the river routing scheme such as in Guimberteau et al. 2012, de Vrese et al. 2018), orwhere the amount of irrigation is limited by using information from observed river runoff
A: For "explicit water amount", a volumetric estimation of water used for irrigation is intended (in this case: VI). This quantity is crucial, as the reviewer correctly pointed out later in the comments, as the impact of irrigation might differ between models or simulations despite identical assumptions. Thus, the importance of this quantity, which is directly related to irrigation and that can be used to compare studies. Irrigation water volume can also be used to compare country-wise estimations, which are independent from the atmospheric/soil models. The lack of this estimation is considered a limitation of the reliability of the irrigation impacts and its magnitude from different studies (Sack et al,2009 and Wada et al, 2013).

3. C: Does this mean that (as irrigation water is added to precipitation) also the actual precipitation is not intercepted? This would introduce an erroneous model change that makes the comparison to the control simulation invalid as effects on model results are not only caused by irrigation itself.
A: In the CHANNEL method, the water coming from the irrigation only does not interact with the canopy. The rain produced by the atmospheric processes does interact with the canopy normally. The sentence is changed in the new manuscript to clarify this point.

4. C: Why a non-reproducible option is given as default, and not the reproducible one? This makes a potential code debugging more difficult.

A: The term non-reproducible is associated to across-compilers, which is related to the random number generator that might be architecture-dependant. While a random number option ensures that the resulting field is randomly distributed, the non-random option ensures that such field has no specific spatial pattern. Neither option is the default, the default is the synchronous activation.

5. C: eq. 1 and line 127-128, Table 2 I don't understand the definition of .Why a frequency is characterized with the difference symbol ? Why a frequency is expressed in number of days and not number per days? If irrigation is conducted once per weak, I would expect a frequency of 1/7 days, and not 7 days such as it is defined.

A: We have changed the term frequency to interval to avoid this confusion.

6. p. 21 – Fig. 11 and 12Figures are too busy. Showing 28 different curves, the curves are not distinguishable.

Fig. 11 has been adapted to highlight better the runs. New figure (Fig. 1, at the end of the document) and the old one (Fig.2, at the end of the document) are here shown. However, it is important to show how all the different timing options behave with respect to the control run. The shading of Fig. 12 and 11 (left) is crucial to highlight the spatial variability.

7. p. 22, 23, 24 – Fig. 13 (upper panels) and 14, 15 (upper panels) Figures are too busy and blurry with all these curves. The light-dashed lines do not provide any additive value and strongly distort the figures.

The upper panels dashed lines are included to highlight the fact that, while the differences with respect to the control runs increase with time, the differences between timing do not. This implies that irrigation itself influences the physical quantities beyond the diurnal timescale of its application. However, the timing

of irrigation does affect these atmospheric/soil variables only within the diurnal cycle.

We thank the reviewer for useful comments and suggestions for further references. The minor remarks corrections are made to the new manuscript.
**R2:**

1. C: add the secondary effects in the Introduction.
A: the introduction has been modified accordingly.

2. C: How does the development here differ from the paper by Lawston et al., 2015, J.Hydrometeor., 1135–1154
First of all, all the schemes in Lawston differs in irrigation timing, frequency and type of water application. Our schemes differ only in the type of application, making the comparison between methods possible. The drip scheme in Lawson is completely different from the one presented here, as our water is intercepted by the canopy and in the other the evapotranspiration is modified as if there was no soil moisture stress. The only similarity with the sprinkler schemes is that there is an explicit irrigation amount. Lawson affirm that their scheme is not driven by crop-water demand, however, the water application is activated and stopped depending on the root-zone soil moisture availability only. Our scheme fulfills this requirement, as the timing of irrigation and the water applied is defined through user-defined parameters. In our scheme the water is applied to the rain water mixing ratio of the lowest model level, in Lawson it is applied as rain rate, therefore already in the surface driver scheme. Therefore, it would resemble more the DRIP scheme presented here. However, it differs as the method presented in our paper has both timing and water amount controlled by the user. The flood method by Lawston applies the water at the root-zone until the top layer is saturated for 30 minutes. The channel method here applies the water at the surface with a prescribed rate and duration, which are controlled by the user. Generally speaking, the three methods described in Lawson do not include any specific information of total water amount used for the schemes as a priori information or timing, which is not realistic. Regarding the decision of the area to irrigate, Lawston relies on the USGS (which is derived from satellite data from 1992 to 1993) to irrigate the whole grid point or half. In this parameterization development, we

use the global FAO dataset of area equipped for irrigation. This allows the application of irrigation to different land use data sources (e.g. MODIS), irrigation on high-rise vegetation (e.g. orchard) and ad-hoc mask modifications (in case of high resolution information available).

3. C: Does the scheme consider the evaporation of water on the leaves (and so the cooling effect)? Does the temperature of irrigated water matter?
The schemes consider evaporation from the leaves only when the schemes allow water to be intercepted, i.e. for both DRIP and sprinkler. The temperature of the water does not matter in the canopy water equation from Noah, therefore is not accounted. The same happens for any possible difference between droplet and soil temperatures differences in the energy budget. This opens to broader investigations topics that go beyond the scope of these parameterizations, and it would require further work from the Land Surface Modelling communities.

4. C: the central pivot irrigation that is the main method for many parts of framing in United States. In addition, over the central Great plains, underground water is pumped for irrigation, and that water can be 10-20C lower than the surface water. Can the three irrigation schemes be applied to the central pivot irrigation from ground water?
A: The sprinkler scheme can be applied for high resolution studies that model such area, for coarse resolution runs (especially in the vertical levels), the DRIP scheme can be more suitable. However, it is not able to represent the potential effect of the difference in temperature in the water used. This is caused by the fact that the microphysics parameterizations assume that rain water is in immediate equilibrium with the air temperature. While this might be a strong assumption for the irrigation case, it allows the sprinkler scheme to work with the microphysics parameterizations without modifying all of them. The issue brought up here might help defining the path to be taken in future studies.

5. C: Irrigation mask field. The work here is similar as Aegerter et al., 2017 in which MODIS-based USDA irrigation database was used. However, it remains unclear how the fraction/percentage of irrigation in a model gridbox is factored into the Noah Land Surface Model in terms of surface properties for that gridbox as whole.

   The irrigation mask is used only for factoring the irrigation water applied, and it is done at the surface driver level (for both DRIP and CHAN) and at the microphysic driver (for SPRI). Therefore, the irrigation water is passed to Noah (and the other LSM) as for the precipitation. The surface properties are defined solely by the land use categories.

6. C: Does the crop types matter over the irrigated area? Aegerter et al. designate that as irrigated cropland and pasture for CLM. How the albedo, leaf area index orNDVI are specified for crops over the irrigated area in NOAH? Obviously, these are the parameters/questions that the present manuscript is not trying to address, but it is important to be clear about it

   No, the crop types do not matter over the irrigated area. All parameters are defined by the standard land use categories which are used by Noah. While this might not be completely realistic, it is used to exactly quantify the response of the model to irrigation alone.

7. C: What surface type database is used? In Figure 2, there are 12 croplands. Are all these croplands irrigated? Does Noah treat these 12 croplands differently in terms of their albedo, leaf area index or NDVI?

   A: This study uses MODIS land use data which has 21 categories, which is shown by the 20 colors in Fig.2 (plus an unassigned category, which is not shown). In Fig. 2 (right), there is only one cropland category (yellow) but it is the number 12 in MODIS dataset, and the land use table employed in the model. The caption of the figure has been changed to not create any misunderstanding.

8. C: Where does 7mm/day come from? Should there be more irrigation in the early
   stage of growing season?
   The 5.7 mm/day is derived from the Eurostat data (as shown in the text), assum-
   ing a constant application throughout the period. We know this is not realistic,
   but we lack any information about sub-seasonal/monthly water application data.
   Uniform application is used in previous studies that use any static/non prognos-
   tic vegetation. A uniform application helps quantifying the impact of irrigation on
   the model at the zero-order modification. A varying irrigation amount should be
   employed in future studies, with dynamic vegetation, to better capture the impact
   of irrigation and agriculture on the studied areas.

9. C: No irrigation and no crops should be the baseline experiment on top of which
   the irrigation effect can be fully studied.
   The baseline has to be defined depending on the research question that is ad-
   dressing. In this case, we aim to introduce the irrigation parameterizations and
   to show how its usage improves the model. The baseline suggested might be
   more realistic in some of the study regions, e.g. very arid areas. However, first it
   should be assessed whether the agricultural area can still exist if it were rainfed.
   This might be done only through a complex LSM that allow dynamic vegetation
   representation, which is not the scope of the current work. Simulations made of
   the current region with the default WRF-ARW model do not include irrigation (but
   still have agricultural areas), which is not realistic. Also, our baseline choice is in
   agreement to what performed in previous studies as Thiery et al. 2017, Sacks et
   al. 2009, Puma et al. 2010, Saeed et. al 2009 (etc).

10. C: Figure 8. Why the colors are different between legend and bar color?
    The colorbar explicitly shows all the control runs (LR1,LR2,SR) and MODIS data.
    It also report the shading legend used to differentiate the irrigated schemes.

11. C: Figure 14. What is shown here is the difference with respect to the control?

How about the difference with respect to the observed T (averaged over all stations)?

We did not show any validation/comparison with measures at the diurnal scale as we lack any information about irrigation timing.

12. C: The irrigation efficiency does depend on the leaf area. In the early growing season, the crop height is low and leaves are small. The efficiency should be similar. With all the assumptions made, it is questionable if the parametrization schemes here have the fidelity to address the issue of irrigation efficiency. From an economic point of view, farmers use irrigation to grow crops, and so, the irrigation amount is unlikely uniform throughout the growing season (as assumed by the model here).

A: The definition used here for irrigation efficiency is added in Sect. 5.4; in this study it aims only to quantify the water loss (in terms of soil moisture changes) depending on the evaporation processes that the water undergoes. Since both the irrigation parameterizations and the other components involved have limitations, the aim is to understand how important each of these evaporative process is at the convection-permitting scales. No parts of this work aims to tackle the efficiency at the single farm scale. The full impact of irrigation as coupled with vegetation is not addressed here as the model does not have the ability to represent dynamical vegetation. This study, however, might give a starting point for further development in future studies.
* * *
[Figure]

**Fig. 1.**

[Figure]

**Fig. 2.**

---

## Referee Report (RR1)

**Manuscript: Evaluation of three new surface irrigation parameterizations in the WRF-ARW v3.8.1 model: the Po Valley (Italy) case study**

**Major remarks**

The authors have revised and improved the manuscript. In their response, the author stated that one of the main objectives of the paper is model documentation with respect to the three parameterizations implemented into WRF. Consequently, some of my previous remarks were considered as not applicable for the present manuscript. If this is ok for GMD, it is ok for me, too.

However, the quality of the English is still not sufficient for a publication in a scientific journal such as GMD. Moreover, my related major remark in my review of the discussion paper has been ignored by the authors. Hence, I repeat this remark: *One major obstacle in reviewing and understanding the paper is the low quality of the English. Several times, I was not able to understand a sentence. Usually, I also provide some suggestions for text edits and sentence improvements, but the high amount of unclear sentences, wrong gramma and unusual sentence structures let me refrain from doing this. I strongly recommend a careful proof reading from a native English speaker.*

In their reply, the authors stated: "New aspects are also the comparison of the impacts of timing as well as the evaporative loss.". This is an important clarification of the novelty of the used parameterizations and I suggest including this statement into the abstract.

I suggest accepting the paper for publication after the English has been considerably improved and minor revisions conducted.

**Minor remarks**

In the following suggestions for editorial corrections are marked in *Italic*.

Line 5
...ascribed *one* cause *to* the ….

Lines 28, 31, 33, 48-50, 59, 65, 87
Text is exceeding the margins.

Line 67
It is written: "… 90-25%"
Do you mean 25-90%?

Line 69
The *second group includes* ...

Line 82
It is written:
"… allows calibrating the F-Parameter ..."

This is a technical detail, a reader, who is not familiar with the CESM code, will not understand. Please explain more thoroughly that it can be understood without this technical knowledge about the code!

Line 89-90
It is written:
"The main reason … … processes."

I cannot follow this argument as this also applies to GCMs

Line 241
cis (2015) is not included in reference list.

Lines 273-283
This paragraph is partially redundant with the definitions and descriptions in Sect. 2 where they belong. Hence, please remove these definitions / descriptions of the parameterizations from this section 3.2.

Lines 358-359
Bias definition is redundant with the one in lines 349-350. Please remove one of them.

Lines 367-368
Incomplete sentence.

Lines 375
It is written:
"The three stations are in an Alpine valley, which can lead to a different set of model biases, such as the effect of steep terrain."

Steep terrain is not a model bias. I do not get what the authors want to say. This is actually a good example for my major remark about the English.

Lines 412-413
It is written:
"The accumulated value obtained for the MODIS data is aggregated with the control run of Fig.8… "

I do not understand this. How does the MODIS data value is aggregated with the control run to yield an accumulated value? Moreover, what is a control run of Fig. 8? This is another good example for my major remark about the English.

Figure 13 – lower panel
Where is the green curve for DRIP?

Lines 652
It is written:
"…. (it is done by comparing the irrigation timing with themselves)."

I do not understand the meaning of this. Actually, the sentence in brackets may be even obsolete.

Line 680
Reference "NCEP …" is not within the alphabetical order.

In addition, I suggest carefully checking all citations to be consistent with the reference list.

---

## Author Response (AR2)

Answer to the reviewer:

The editor told us that, if accepted, the manuscript will undergo professional review in the copy-editing process. This should help improve the sentence structure and readability of the manuscript. Also, the manuscript has been reviewed by the second author (native English speaker).

We answered the reviewer's comments in the manuscript and applied all the changes. A tracked version is uploaded.

https://www.overleaf.com/5146161575sfwcxwjhfzgg

[revised manuscript text omitted]